# Dynamic ice–ocean pathways along the Transpolar Drift amplify the dispersal of Siberian matter

Georgi Laukert [1,2,3,4,5] ✉, Dorothea Bauch [5,6], Benjamin Rabe [7], Thomas Krumpen [7], Ellen Damm [7], Markus Kienast [4], Ed Hathorne [5], Myriel Vredenborg [7], Sandra Tippenhauer [7], Nils Andersen [6], Hanno Meyer [8], Moein Mellat [8], Alessandra D'Angelo [9], Patric Simões Pereira [10,13], Daiki Nomura [11], Tristan J. Horner [2,3], Katharine Hendry [1,12] & Stephanie S. Kienast [4]

The Transpolar Drift (TPD) plays a crucial role in regulating Arctic climate and ecosystems by transporting fresh water and key substances, such as terrestrial nutrients and pollutants, from the Siberian Shelf across the Arctic Ocean to the North Atlantic. However, year-round observations of the TPD remain scarce, creating significant knowledge gaps regarding the influence of sea ice drift and ocean surface circulation on the transport pathways of Siberian fresh water and associated matter. Using geochemical provenance tracer data collected over a complete seasonal cycle, our study reveals substantial spatiotemporal variability in the dispersal pathways of Siberian matter along the TPD. This variability reflects dynamic shifts in contributions of individual Siberian rivers as they integrate into a large-scale current system, followed by their rapid and extensive redistribution through a combination of seasonal ice–ocean exchanges and divergent ice drift. These findings emphasize the complexity of Arctic ice–ocean transport pathways and highlight the challenges of forecasting their dynamics in light of anticipated changes in sea ice extent, river discharge, and surface circulation patterns.

The Transpolar Drift (TPD) is an important surface current in the Arctic Ocean, transporting fresh water and various types of matter from the Siberian Shelf to the Fram Strait and the Canadian Arctic Archipelago[1,2]. This transport, driven by sea ice drift and surface water advection, encompasses a wide range of dissolved and particulate substances originating from Siberian rivers, including nutrients, metals, gases, carbon, microplastics, and other pollutants[3–8]. The redistribution of this matter across the Arctic and into the North Atlantic impacts the region's marine ecosystems, biogeochemical cycles, food security, and human health[9–11], extending the TPD's influence well beyond the effects of fresh water on large-scale oceanic and climate dynamics[12].

[1]School of Earth Sciences, University of Bristol, Bristol, UK. [2]Department of Marine Chemistry & Geochemistry, Woods Hole Oceanographic Institution, Woods Hole, USA. [3]NIRVANA Labs, Woods Hole Oceanographic Institution, Woods Hole, USA. [4]Department of Oceanography, Dalhousie University, Halifax, Canada. [5]GEOMAR Helmholtz Centre for Ocean Research, Kiel, Germany. [6]Christian-Albrechts University of Kiel, Kiel, Germany. [7]Alfred-Wegener-Institut, Helmholtz-Zentrum für Polar- und Meeresforschung, Bremerhaven, Germany. [8]Alfred-Wegener-Institut, Helmholtz-Zentrum für Polar- und Meeresforschung, Potsdam, Germany. [9]University of Rhode Island, Graduate School of Oceanography, Narragansett, USA. [10]Department of Marine Sciences, University of Gothenburg, Gothenburg, Sweden. [11]Hokkaido University, Hakodate, Japan. [12]British Antarctic Survey, High Cross, Madingley Road, Cambridge, UK. [13]Present address: Institute of Geology, University of Innsbruck, Innsbruck, Austria. ✉e-mail: georgi.laukert@bristol.ac.uk

Despite their importance, the mechanisms governing the transport of Siberian matter along the TPD remain poorly understood. A key challenge is distinguishing the respective contributions of sea ice drift and ocean surface circulation to matter transport. Accurate predictions of matter redistribution depend on a detailed understanding of these transport mechanisms alongside focused research into the physical properties and biogeochemical processes that independently influence matter movement in ice and water. Sea ice drift within the TPD is largely wind-driven, whereas surface circulation is shaped by a combination of ice–ocean drag and Ekman transport[13]. Additional circulation features—such as geostrophic currents, topographically constrained boundary currents and their detachments, upwelling between ice floes, and various scales of eddies—further modulate surface water transport and, in turn, ice motion through reverse ice–ocean drag[14]. Chemical exchanges between sea ice and the surface ocean during ice formation, drift, and melting add another layer of complexity, creating seasonally varying ice–ocean transport pathways. Current observational tools and models face limitations in resolving these pathways. Satellite data and drifting buoys provide valuable insights into sea ice origin and movement[15] but offer only a partial understanding of surface hydrography. Numerical models hold promise for integrating sea ice and the surface ocean as interconnected matter reservoirs, but they require fine-scale resolution[16] and precise representation of atmosphere–ice–ocean momentum transfer[17,18]. While some models isolate specific transport mechanisms effectively[19,20], they often fall short in capturing the full extent of matter redistribution[21]. Ship-based observations, although invaluable for studying ice–ocean interactions and tracking matter transport, are limited by the logistical difficulties of accessing the central Arctic, particularly in winter. Overcoming these limitations calls for year-round monitoring with advanced observational tools capable of capturing the dynamic processes in and between sea ice and surface water that govern matter dispersal.

Source-sensitive geochemical tracers offer a promising approach for identifying large-scale matter dispersal trends by integrating circulation effects over time while smoothing out short-term, small-scale variability. Recent advancements in the combined use of stable oxygen isotopes ($\delta^{18}O$), radiogenic neodymium isotopes ($\varepsilon_{Nd}$) and rare earth element (REE) concentrations in seawater have significantly improved our ability to trace fresh water and matter in the Arctic Ocean[22–25]. These tracers are particularly effective in the Eurasian Basin, where processes that could alter dissolved $\varepsilon_{Nd}$ and REE distributions—such as seawater–particle exchange[26]—can be accounted for or excluded. Studies employing these tracers have revealed substantial variability in contributions from individual Siberian rivers across the Siberian Shelf[24] and the Eurasian Basin[25]. Fluctuations in micro- and macronutrients are linked to these contributions[27,28], underscoring their ecological importance. The application of these provenance tracers to sea ice[29,30] has further enhanced understanding of ice–ocean interactions and Arctic transport processes.

This study applies these tracers to explore the dispersal of Siberian matter by sea ice drift and ocean surface circulation along the TPD throughout the annual cycle. Building on previous Arctic provenance tracer studies, we quantitatively decompose water masses using seasonal water column and sea ice data from the Multidisciplinary drifting Observatory for the Study of the Arctic Climate (MOSAiC) expedition conducted from October 2019 to September 2020 ('Methods'). Our analysis focuses on polar winter, examining the signatures of sea ice and surface seawater over a ~200-day drift (Fig. 1). Observations from MOSAiC revealed considerable hydrographic variability beneath the sea ice, driven mainly by surface circulation and seasonal atmospheric forcing[31–33]. These findings are especially critical in light of the recent acceleration of sea ice drift[34,35] and the sharp reductions in sea ice extent and thickness due to Arctic warming[36]. These shifts overlay interannual and fine-scale fluctuations in flow structure[16] and broader variability tied to the Arctic Oscillation[37].

Understanding the mechanisms driving matter transport along and beyond the TPD is essential for assessing natural variability and predicting the future impacts of climate change.

## Results and discussion

### Context of the MOSAiC campaign: Sea ice drift and ocean surface hydrography

The MOSAiC drift campaign began in October 2019 when the *RV Polarstern* moored to an ice floe at 85° N and 137° E. This floe, formed in December 2018 near the New Siberian Islands[38], served as the vessel's drifting platform until it disintegrated in July 2020. The ship then relocated to the central Arctic Ocean, continuing its drift with a new floe through August and September 2020. The original floe, which survived the 2019 summer melt and was classified as second-year ice on January 1, 2020[39], was first sampled in October 2019. Shortly before, between September 1–13, 2019, first-year ice began forming in open waters or beneath the second-year ice as insulated first-year ice[40]. Figure 1 depicts the ice formation areas, floe drift paths, and sampling locations.

Throughout the campaign, dynamic oceanographic conditions were apparent. Surface current velocities averaged 3.3 km day$^{-1}$ in winter 2019/2020, increasing to 4.3 km day$^{-1}$ in August/September 2020 after *RV Polarstern* relocated to the central Arctic Ocean[32]. Despite these averages, fine-scale variations were significant, with peak speeds in the upper 50 m reaching 22 km day$^{-1}$ during early spring 2020. Ice drift velocities frequently exceeded surface current velocities, with the predominant ice drift direction often diverging from the underlying water movement[31,33].

Rabe, et al.[31,32] and Schulz, et al.[33] provide detailed analyses of hydrography during the MOSAiC drift campaign, summarized here with an emphasis on the upper water column. Throughout the drift, Arctic Atlantic Water (AAW)—a colder, less saline form of Atlantic Water—was observed at depths of 200–400 m. Above this layer, a stable halocline was present from October 2019 through March/April 2020, after which the mixed layer began to deepen. In the upper 40 m, near-freezing temperatures prevailed through winter 2019/2020 until May/June 2020. As the drift progressed into the southern Nansen Basin, mixed layer temperatures began to rise. Surface salinity had started to increase earlier, during the transition from the Amundsen Basin to the Nansen Basin, shifting from below 32 in early 2020 to above 34 by midyear. Beyond early 2020, low surface salinities and temperatures were detected in the Fram Strait in late July 2020, during *RV Polarstern's* transit to the second ice floe in August 2020, and near the North Pole during the second drift in late August and September 2020 (Fig. 2a). Examining the origins and characteristics of these low-salinity waters—defined here as having salinities ≤32—is a central focus of this study.

### Origin, distribution and variability of fresh water along the Transpolar Drift

Fresh water in the surface layer of the Eurasian Basin primarily originates from the Siberian Shelf, with substantial contributions from river runoff[7,8]. After crossing the shelf and mixing to varying degrees with AAW, this water integrates into the TPD. The extensive $\delta^{18}O$ datasets[41,42] reveal a robust positive correlation between $\delta^{18}O$ and salinity in the upper water column ( <250 m, $R^2 = 0.85$, $p < 0.05$; $\delta^{18}O$ and salinity sections shown in Supplementary Fig. S1), indicating that river water and AAW were the dominant contributors to the mixed layer and halocline during the MOSAiC drift. Using these datasets and a three-component water mass analysis based on $\delta^{18}O$ and salinity ('Methods'), we mapped the high-resolution distribution of river water content ($f_{RIV}$) along the MOSAiC drift (Figs. 2b, 3a). Surface $f_{RIV}$ reached approximately 21% near the North Pole in late August and September 2020, confirming a significant riverine contribution to the low-salinity waters. Elevated $f_{RIV}$ values, peaking at 15%, were also observed nearby during January and February 2020, in Fram Strait by late July 2020, and between Fram Strait and the North Pole during *RV Polarstern's* transit

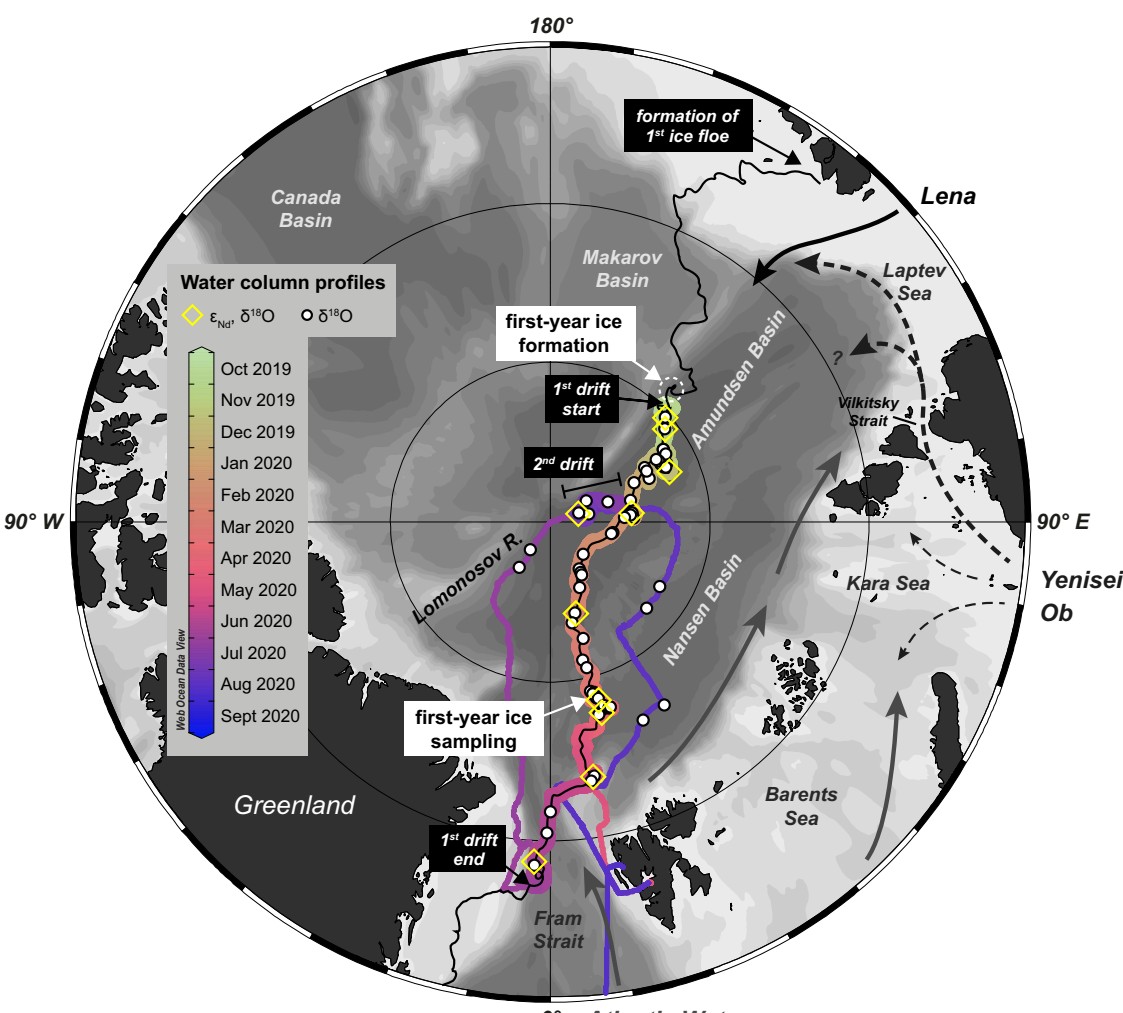

**Fig. 1 | Map of the MOSAiC expedition drift tracks and locations of provenance tracer sampling.** The paths of the 1st and 2nd drifts (depicted by thick lines) and the transfers between them (shown as thin lines) are color-coded by date. In addition to water column stations, the start and end points of both drifts and the areas of first-year ice formation and sampling are highlighted. Since the exact location of first-year ice formation is unknown, a dashed white circle marks the drift interval during which first-year ice formation likely occurred (see main text for more information). The trajectory of the 1st ice floe, both prior to the start of the 1st MOSAiC drift in October 2019[38] and after its conclusion in September 2020, is depicted by a thin, continuous black line. The source region of this ice floe is also marked. The approximate advective freshwater pathways on the Siberian Shelf are represented by a solid black arrow for the input from the Lena river and by several dashed black arrows for the contributions from the Yenisei and Ob rivers. The inflow of Atlantic Water through Fram Strait and the Barents Sea, and its subsequent route along the Arctic boundary current are indicated by solid grey arrows. The map was created using ODV[82] and modified manually.

to the second ice floe in August 2020 (Fig. 2b). These findings, alongside persistently low sea surface salinities during the transit, suggest that the low-salinity waters between the Lomonosov Ridge and the western Fram Strait were interconnected components of a larger, continuous flow of river water across the Arctic. In contrast, in the Nansen Basin from April to June 2020, surface $f_{RIV}$ dropped to approximately 5%, reflecting a stronger influence of AAW on the mixed layer and halocline, consistent with the low stratification observed in this region[32,33]. Apart from thin surface meltwater lenses (<1 m) formed locally during summer 2020[43], contributions from sea ice meltwater ($f_{SIM}$) were generally low throughout the campaign, with fractions below 2% beneath the halocline. However, brine contributions (negative $f_{SIM}$) reached up to 10% in the mixed layer and halocline over extended periods. These brines likely originated partially from productive polynya regions on the Siberian Shelf, as sea ice formation along the TPD alone cannot fully account for their presence. Although a comprehensive analysis of the $f_{SIM}$ distribution is beyond the scope of this study, we observe that its patterns closely align with those of $f_{RIV}$ and river-borne nutrients, such as silicic acid[44]. This similarity

underscores the strong compositional link between the Siberian Shelf and the Eurasian Basin throughout the MOSAiC drift.

Dissolved $\varepsilon_{Nd}$ compositions and REE concentrations offer deeper insights into the origin of river water and its interactions with AAW (Figs. 2c and 3b, and Supplementary Figs. S1 and S2). Neodymium isotopes are particularly effective tracers for ocean circulation due to the quasi-conservative behavior of Nd in the oceans and its long average residence time of several hundred years[45]. Even in surface waters with shorter residence times, Nd isotopes can effectively track short-term water mass advection and mixing. In the TPD, where ice and water transport occur on timescales shorter than a decade[3,33,46], non-conservative processes, such as boundary exchange[26], could alter dissolved $\varepsilon_{Nd}$ signatures over shorter periods. However, both our data and summer data from 2015[25] suggest that such processes have little influence on the dissolved $\varepsilon_{Nd}$ distribution in the upper water column of the Eurasian Basin (Supplementary Text 1). Instead, $\varepsilon_{Nd}$ signatures and REE concentrations primarily reflect mixing between AAW and waters from the Lena, Yenisei, and Ob rivers, aside from one outlier sample from the Fram Strait likely influenced by Greenland inputs. Compared to the 2015

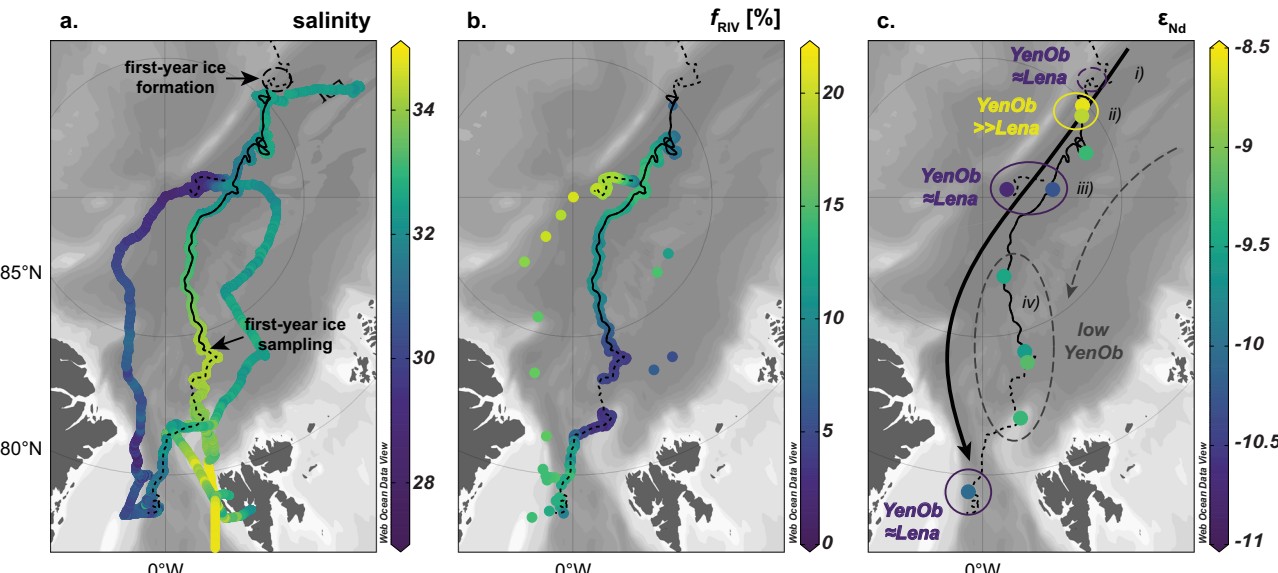

**Fig. 2 | Surface distribution of salinity, river water fraction and neodymium isotopes along the TPD. a** Continuous salinity measurements taken at 11 m depth using two thermosalinographs (SBE21, SeaBird GmbH) on board the *RV Polarstern* during both drifts and transit to and from the second ice floe[83–87]. **b** River water content ($f_{RIV}$) in percent derived from the extensive stable oxygen isotope datasets[41,42]. **c** Seawater neodymium isotope composition ($\varepsilon_{Nd}$)[88], with major compositional trends depicted by colored circles that reflect the relative proportions of the two riverine endmembers Lena and Yenisei/Ob (YenOb > >Lena: pronounced excess of Yenisei/Ob water compared to Lena water; YenOb≈Lena: nearly equal contributions of these endmembers; see also Figs. 3e and 4b for interpretation). The dashed black circle in (**a**) marks the formation region of first-year ice, for which no seawater data are available. However, river water composition in this region, indicated in (**c**), is inferred from the first-year ice data. The pathway of the freshwater-rich Transpolar Drift, showing fluctuating river water contributions, is represented by a thick, solid black arrow in (**c**), while the advection of low amounts of Yenisei/Ob water along the Nansen Basin, likely exported from the northwestern Laptev Sea, is shown in the same map with a dashed grey arrow and a grey dashed circle around relevant stations. The drift tracks of the ice floes are shown as dashed black lines in all maps, while the recorded growth and drift of first-year ice are represented by a solid black line. The intervals i)–iv) shown in **c** correspond to major trends in river water proportions in the sea ice. The maps were created using ODV[82] and modified manually.

dataset, which included the Makarov Basin, the MOSAiC dataset—more focused on the TPD—reveals an even stronger correlation between $f_{RIV}$ and [Nd] ($R^2 = 0.91$, $p < 0.05$, Fig. 4a). This relationship, along with progressive $\varepsilon_{Nd}$ shifts with increasing $f_{RIV}$ and decreasing depth (Fig. 4b), underscores the dominant role of these rivers and AAW in supplying dissolved REEs to the Eurasian Basin. The $\varepsilon_{Nd}$ shifts reflect gradual admixture of river water to AAW, initially from the Yenisei and Ob rivers ($\varepsilon_{Nd} = -6.1$ to $-5.2$) and subsequently from the Lena river ($\varepsilon_{Nd} = -16.7$ to $-15$). Each mixing step introduces a distinct $\varepsilon_{Nd}$ signature, while the similar REE concentrations among the rivers sustain a consistent increase in [Nd] irrespective of their relative contributions. This understanding supports an expanded four-component water mass model that integrates $\varepsilon_{Nd}$ with $\delta^{18}O$ and salinity to distinguish contributions from the Lena ($f_{Lena}$) and Yenisei/Ob ($f_{YenOb}$) rivers ('Methods'). Adjusted REE concentrations for these endmembers account for estuarine processes that reduce riverine REE concentrations before entering the TPD (Supplementary Text 2.2). The analysis indicates that waters with the highest $f_{RIV}$ exhibited nearly equal contributions from the Lena and Yenisei/Ob rivers (Fig. 3c, d). However, their proportions varied considerably during the drift, with individual contributions ranging from 0% to 9–10%. Furthermore, elevated Lena contributions were consistently accompanied by at least ~6% Yenisei/Ob water. The excess of Yenisei/Ob water over Lena water ($f_{YenOb\_ex}$) was most pronounced at the surface during the early phase of the first drift (October/November 2019) and at subsurface depths (50–100 m) in early 2020 and September 2020. Nearly equal contributions were observed at the surface in January, late July, and September 2020, thus particularly in the presence of low-salinity waters (Figs. 2c and 3e). The admixture of significant AAW volumes to the mixed layer and halocline in the Nansen Basin, inferred from low surface $f_{RIV}$ values between April and June 2020, is reflected in shifts in $\varepsilon_{Nd}$ compositions and REEs towards AAW core values ($\varepsilon_{Nd} \sim -11$, [Nd] $\sim$16 pmol kg$^{-1}$), consistent with previously

reported AAW characteristics in the Eurasian Arctic Ocean[22–25]. While definitive tracers for Pacific water remain unavailable, our data and recent studies suggest negligible Pacific water influence in the Eurasian Basin (Supplementary Text 2.4), supporting earlier estimates[25].

The predominance of Yenisei/Ob water over Lena water ($f_{YenOb\_ex} > 0\%$) aligns with established shelf circulation patterns[46–48] and previous tracer observations in the Eurasian Arctic Ocean[22–25]. The transformation of Atlantic Water into AAW, occurring through gradual mixing with Siberian runoff along the shelf and boundary currents, generates low salinity shelf waters in the Barents, Kara and Laptev seas, which subsequently disperse into the Eurasian Basin[47]. During the MOSAiC drift in October and November 2019, the mixed layer was primarily composed of AAW and fresh water from the Yenisei and Ob rivers, with surface $f_{YenOb}$ peaking at 8% (Fig. 3c). This two-component mixture likely originated in the Kara Sea, where AAW first interacts with substantial Arctic river runoff. Before joining the TPD, this Kara Sea water is likely advected into the Laptev Sea through the Vilkitsky Strait Current. Although the Yenisei/Ob plume dominates the central Kara Sea during summer and fall, it does not directly enter the Eurasian Basin. Instead, the plume migrates into the northwestern Laptev Sea during winter and largely vanishes from the Kara Sea by early spring[49]. By September, surface $f_{YenOb}$ can still reach up to 19% in the northwestern Laptev Sea[50], highlighting this pathway's significance even in summer. In the Laptev Sea, this water mixes variably with Lena water, forming high-$f_{RIV}$ surface waters that are exported north into the TPD. Denser Kara Sea waters, characterized by lower $f_{YenOb}$ due to stronger mixing with AAW, submerge beneath these high-$f_{RIV}$ waters whenever the latter are present. This dynamic creates a distinct shelf-derived stratification feature within the TPD, observed in 2015[25] and during the MOSAiC drift in early 2020 and again in September of the same year (Fig. 3e). Direct mixing between AAW and Lena water is unlikely along this route, as AAW first interacts with Yenisei/Ob waters before

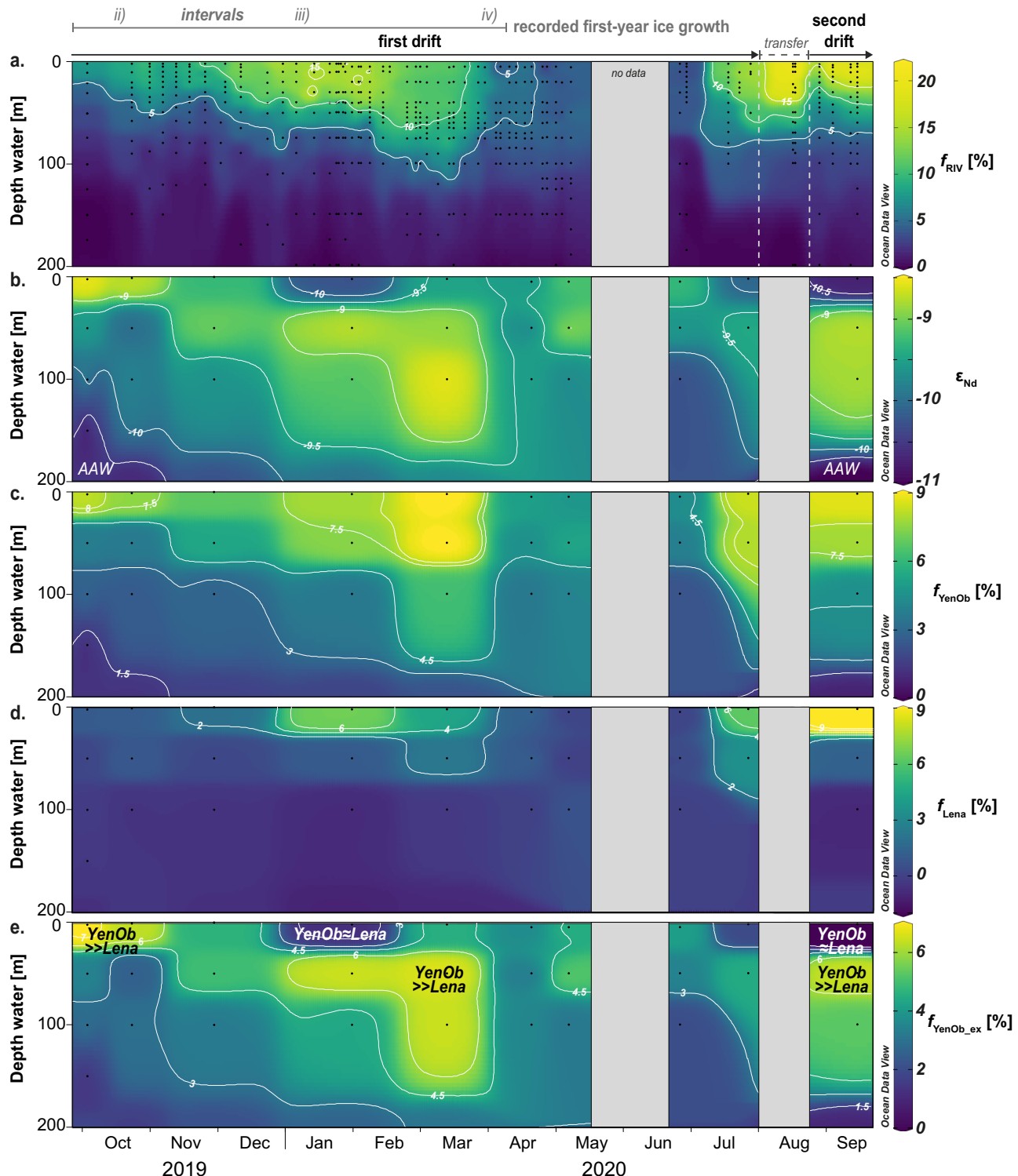

**Fig. 3 | Evolution of provenance tracer parameters in the upper water column along the MOSAiC drift track. a** River water content ($f_{RIV}$, in percent) derived from the extensive stable oxygen isotope ($\delta^{18}O$) dataset[41] (original salinity and $\delta^{18}O$ data are shown in Supplementary Fig. S1). **b** Seawater neodymium isotope composition ($\varepsilon_{Nd}$)[88] (corresponding neodymium concentrations shown in Supplementary Fig. S1). **c** Contributions of the Yenisei and Ob rivers ($f_{YenOb}$, in percent). **d** Contributions of the Lena river ($f_{Lena}$, in percent). **e** Excess of Yenisei/Ob river water relative to that of the Lena river ($f_{YenOb\_ex}$, in percent), with marked zones of a pronounced excess of Yenisei/Ob water over Lena water (YenOb >>Lena) and nearly equal contributions of these endmembers (YenOb≈Lena). The grey line indicates recorded first-year ice growth along the drift, with an approximate temporal assignment of sea ice intervals ii)–iv). Grey areas indicate transit periods when the *RV Polarstern* traversed between ice floes and land, resulting in no data collection except for $\delta^{18}O$ at two stations with shallow sampling during transit to the 2nd ice floe (see Fig. 1). The sections were created using ODV[82] and modified manually.

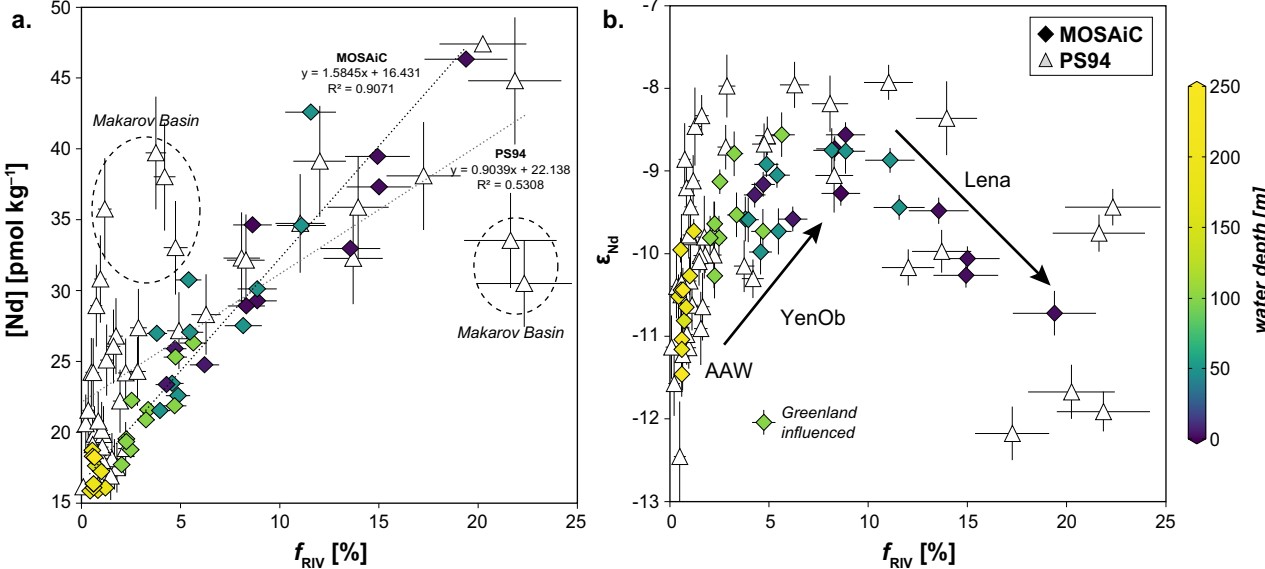

**Fig. 4 | Comparison between provenance tracer data obtained from the MOSAiC drift[88] and the PS94 cruise (GEOTRACES transect GN04, 2015[25]).**
**a** River water content ($f_{RIV}$, in percent) derived from the three-component analysis plotted against neodymium concentration ([Nd], in pmol kg$^{-1}$). **b** River water fraction plotted against neodymium isotope composition ($\varepsilon_{Nd}$). Error bars for $\varepsilon_{Nd}$ and [Nd] represent ± 2 standard deviations from measurements, while those for $f_{RIV}$ account for both measurement errors and endmember variability. Black arrows

indicate admixture of individual riverine Nd sources (Lena, YenOb) to Arctic Atlantic Water (AAW). PS94 cruise samples collected in the Makarov Basin, outside the freshwater-rich part of the Transpolar Drift, are highlighted with dashed circles. One MOSAiC sample from the Fram Strait, influenced by unradiogenic $\varepsilon_{Nd}$ from Greenland (Supplementary Text 2), deviates from the data envelope in the $f_{RIV}$-$\varepsilon_{Nd}$ space and is excluded from our analysis. MOSAiC data symbols are color-coded by sampling depth.

reaching the Laptev Sea. This explains the absence of pure AAW-Lena mixtures during the MOSAiC drift. Nevertheless, Lena river water distinctly influences the water composition, evidenced not only by shifting $\varepsilon_{Nd}$ signatures but also flatter REE patterns (Supplementary Fig. 2, normalized to PAAS, Post-Archaean Australian Shale[51]) and thus lower PAAS-normalized heavy-to-light REE ratios (HREE/LREE) observed in the low-salinity waters with the highest $f_{Lena}$. Such REE characteristics stem from interactions with weathering inputs either in the river catchments or at the land-ocean interface[24,52]. Rapid transport of Lena water into the Eurasian Basin partly preserves these patterns within the TPD, countering oceanic REE fractionation. This fractionation occurs during estuarine REE removal through preferential scavenging of LREEs upon mixing with seawater[53], gradually increasing HREE/LREE ratios to levels typical of AAW[24].

The river water distribution along the TPD during the MOSAiC expedition reveals notable spatiotemporal variability in advection. Variations in ice drift direction caused by inertial oscillations[54]—the circular movement of sea ice driven by the Coriolis effect—appear to have no impact on tracer-derived fraction distributions when plotted over time and thus following ice motion, likely due to the low spatial resolution of the tracer data. Even in the well-resolved river water distribution, only broad advection and mixing patterns are visible, while smaller features fall below data resolution (Fig. 3a). These patterns also remained unaffected by local ice–water interactions during the drift, as tracer-derived fractions are consistent throughout the upper water column, extending to depths of up to 100 meters—well beyond the direct influence of ice–ocean drag and Ekman transport. Throughout the campaign, sea ice drifted up to six times faster than surface waters[33], moving over high-$f_{RIV}$ waters toward the Fram Strait until mid-March 2020, when the drift shifted towards the Nansen Basin. Therefore, tracer changes observed until spring 2020 mainly reflect variability in river water advection along the freshwater-rich TPD, along with the high-$f_{RIV}$ waters found later that year near the North Pole and at Fram Strait. The nearly equal contributions from the Lena and Yenisei/Ob rivers at these two sites, as well as during winter 2020, suggest a relatively stable river

water composition over a distance of approximately 1500 km, spanning from the Lomonosov Ridge to western Fram Strait (Fig. 2a–c). However, sustaining such a stable river water composition along the TPD faces two significant challenges. First, strong variability in river water composition is evident before the appearance of the first low-salinity water in early 2020. The MOSAiC floe likely drifted through a Lena-Yenisei/Ob mix for about a month before the *RV Polarstern* arrived in October 2019, as indicated by first-year ice data (see next section). Over the seven-month drift spanning 600 km, river water composition thus fluctuated from a Lena-Yenisei/Ob mixture to predominantly Yenisei/Ob waters and back to a mixed composition (Fig. 2c). Second, strong seasonal and interannual hydrographic shifts on the Siberian Shelf, particularly in the Laptev Sea, make long-term export of a constant river water composition highly unlikely. A consistent composition between the Lomonosov Ridge and western Fram Strait would require continuous export of constant Lena-Yenisei/Ob mixtures over one to four years, assuming surface water velocities between 1 and 5 km day$^{-1}$ [7,32,55]. This scenario is unlikely, as nearly half of Siberian river runoff occurs within a two-month period around the spring freshet, producing river-influenced regions with notable seasonal and interannual variability[49,50]. Wind patterns further influence the Lena plume's direction, pushing it north into the TPD or eastward into the East Siberian Sea. Yenisei/Ob waters, variably advected from the Kara Sea[49], occupy the central shelf when the Lena plume is deflected eastward[24,50], likely experiencing similar export mechanisms into the TPD. The variability observed during the first half of the MOSAiC drift suggests that distinct river water mixtures exited the shelf within 40 to 200 days (based on above noted surface water velocities), aligning with surface water residence times of weeks to months in the Laptev Sea[56]. This suggests that hydrographic shifts on the shelf precondition the TPD, driving temporal variability in surface water composition across the entire freshwater-rich TPD, not just the 600 km segment where riverine contributions are distinguishable.

The simultaneous emergence and advection of pure Yenisei/Ob-AAW mixtures from the northwestern Laptev Sea likely explain the low $f_{YenOb}$ values of up to 5% observed in the Nansen Basin between April

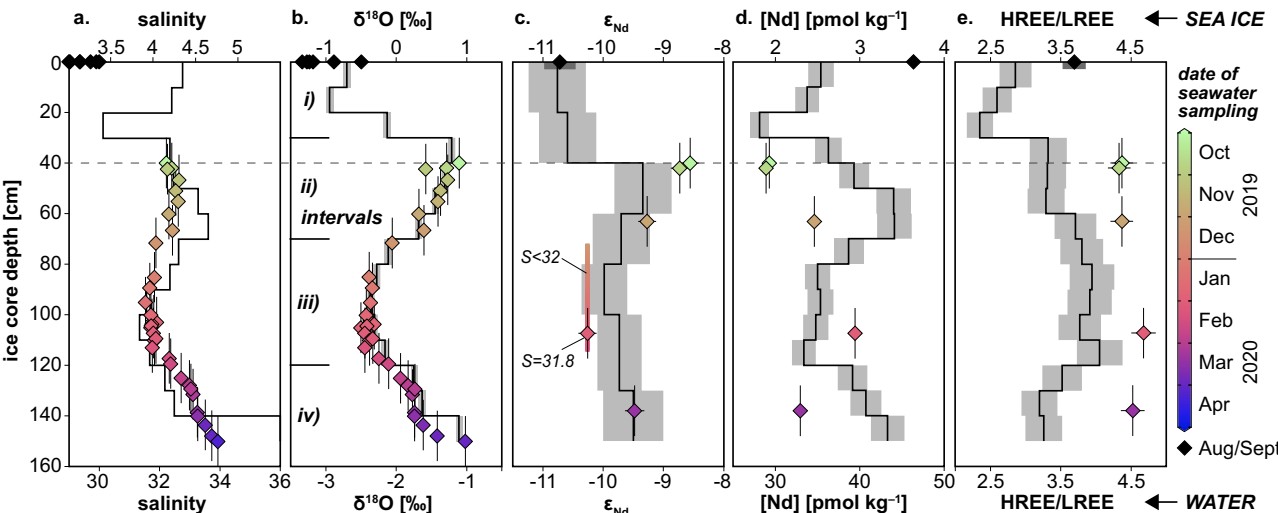

**Fig. 5 | Comparison between first-year ice profiles[89] (upper x-axes) and surface seawater composition[41,42,88] (lower x-axes). a** Salinity (bulk salinity for sea ice). **b** Stable oxygen isotope compositions ($\delta^{18}O$). **c** Neodymium isotope compositions ($\epsilon_{Nd}$). **d** Neodymium concentrations ([Nd]), and **e** Post-Archean Australian Shale normalized[61] heavy-to-light rare earth element ratios (HREE/LREE). The first-year ice, recovered on April 8, 2020, is represented by tracer profiles derived from nine individual ice cores, with depth horizons merged to generate the presented data. In contrast, surface seawater data were collected throughout the ice growth and drift period but are plotted in the first-year ice profiles at ice core depths corresponding to the seawater sampling dates. These depths were calculated using an age model for the sea ice ('Methods'). The sea ice composition is depicted by black lines, representing average values across varying depth segments, with grey fields indicating 2 standard deviations from measurements. Surface seawater composition, sampled at depths of 2 to 5 m, is represented by color-coded diamonds indicating the sampling date. For the upper 40 cm, where seawater data are unavailable, black diamonds denote low-salinity surface waters in August/September 2020 near the North Pole. Dark grey fields in (**c** and **e**) for the latter and x-axis error bars for the other seawater data represent 2 standard deviations from measurements for the $\epsilon_{Nd}$ and HREE/LREE values. Y-axis error bars on seawater data indicate ice age uncertainty. In panel (**c**), a color-coded line represents the ice core depth interval corresponding to ice drift when surface seawater salinities were below 32 (December 12, 2019, to February 10, 2020). Depth ranges for the four distinct sea ice intervals i)–iv) are indicated in the $\delta^{18}O$ profile in (**b**). Differences in absolute values between sea ice and seawater composition arise from brine rejection for salinity, [Nd] and HREE/LREE ratios, and from fractionation during sea ice formation for $\delta^{18}O$.

and June 2020 (Figs. 2c and 3c). However, this advection appears confined to the Nansen Basin and is unlikely linked to the variability seen in the freshwater-rich segment in the Amundsen Basin. Strong compositional fluctuations along the 600 km segment—occurring without significant changes in ice drift direction—suggest that temporally variable export is the driving factor of these compositional changes, rather than the floe crossing distinct but relatively stable contributions from the western and eastern Laptev Sea. Basin circulation also seems to have had minimal influence on this distribution, as no cross-currents or large-scale features were detected[33]. Moreover, the low occurrence of eddies observed during MOSAiC[33] suggests limited impact of mid-scale phenomena on large-scale water mass distribution. Instead, river water transport along the TPD is primarily governed by the mean geostrophic flow, which is influenced by wind- and ice-induced surface stress affecting sea surface height. Although short-term variability in all these components can affect matter transport, the mean transport aligns with this large-scale flow, which is fed by headwaters from the eastern Laptev Sea with shifting river water content and composition. This evolving understanding of ocean surface variability along the TPD emphasizes the critical role of shelf hydrography and seasonality in shaping freshwater transport, with important implications for the composition of sea ice formed along the drift route and the associated redistribution of Siberian matter.

### Uptake of Siberian matter by sea ice throughout ice growth and drift

A discernible and consistent pattern emerges in the profiles of all tracer parameters and bulk salinity within first-year ice, indicating that surface water incorporation is the primary mechanism influencing their distributions (Fig. 5). We identified four depth intervals based on these variations: i) 0–30 cm, ii) 30–70 cm, iii) 70–120 cm, and iv) 120–150 cm (Fig. 5b). The $\delta^{18}O$ profile best illustrates these intervals,

with values as low as −0.95‰ in intervals i) and iii), contrasting with higher values of up to +0.89‰ in intervals ii) and iv) (Fig. 5). Bulk salinity, [Nd], $\epsilon_{Nd}$ and the HREE/LREE ratio also exhibit these intervals, though shifted downwards by about 10 cm. This shift likely results from the sensitivity of these parameters to permeability changes during sea ice maturation (Supplementary Text 3). While $\delta^{18}O$ integrates largely into the ice crystal lattice, major and trace elements, including salts and REEs, are excluded and accumulate in interstitial brine channels and pockets[57]. This brine rejection lowers salt and REE concentrations relative to parental seawater and, along with small-scale brine heterogeneity, results in a more varied distribution within the sea ice (Supplementary Text 3). By aggregating corresponding ice core depth segments across multiple cores ('Methods'), we minimized the impacts of brine heterogeneity and brine spillage during sampling, isolating the effects of permeability changes and brine rejection. Notably, these factors do not appear to alter the influence of parental seawater composition, even for steady-state salinity, despite internal temperature- and salinity-induced changes affecting ice porosity and microstructure[57]. Typical C-shaped profiles of salinity were observed during the initial growth phase of the first-year ice but were replaced by less variable profiles later in the winter[39]. These profiles provide a timeline based on numerous individual ice cores, although none captured the finer fluctuations in salinity distribution reported here for the same period. Future studies exploring the influence of surface seawater composition on salinity and brine-associated substances in sea ice may find value in a sampling strategy like ours, which effectively reduces the influence of small-scale variability caused by changes in porosity.

A significant outcome of the MOSAiC drift campaign is the direct comparison of tracer signatures in the first-year ice with those of parental surface seawater from which the ice formed. Continuous sea ice thickness measurements and regular ocean surface sampling enable a

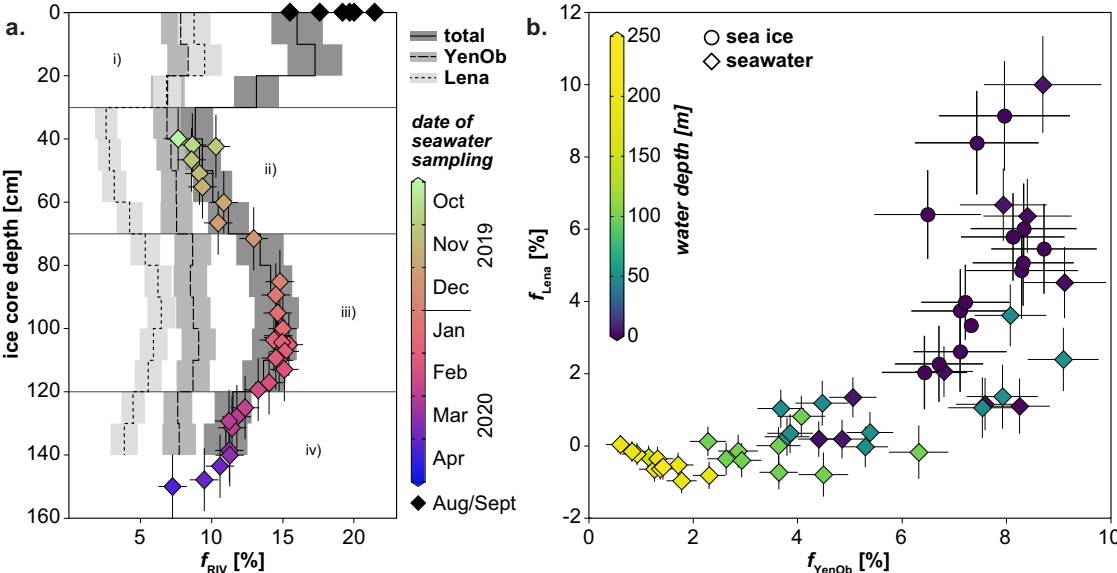

**Fig. 6 | Relationship between individual riverine contributions in surface seawater and sea ice. a** Distribution of the percentage contributions of Lena and Yenisei/Ob rivers (Lena, YenOb) within the sea ice profile and their link to surface seawater values. The latter are represented by color-coded and black diamonds that indicate sampling dates, following the same scheme as in Fig. 5. The total riverine contribution in sea ice ('total' = $f_{Lena} + f_{YenOb}$) closely corresponds to $f_{RIV}$ determined for surface seawater samples via the three-component analysis.

**b** Comparison of the percentage contributions from the two riverine endmembers to surface seawater (diamonds) and sea ice (circles). Symbols are color-coded to represent sampling depth. Error bars in (**a**, **b**) represent uncertainties in water mass assessment related to measurement inaccuracies and endmember variability, except for y-axis error bars on seawater data in (**a**), which represent ice age uncertainty.

direct linkage of surface seawater composition to specific sea ice depth horizons ('Methods'). Notably, there is excellent agreement between first-year ice and surface seawater values for salinity, $\delta^{18}O$, and $\varepsilon_{Nd}$, despite differences in absolute salinities and $\delta^{18}O$ values (Fig. 5). As discussed previously, brine rejection influences salt content, leading to lower bulk salinity in sea ice. The observed offset of ~2 ‰ in $\delta^{18}O$ between seawater and sea ice matches previous estimates of oxygen isotope fractionation during sea ice growth[8], suggesting a consistent fractionation process during first-year ice formation ('Methods'). Minor discrepancies in tracer profiles between ice and water likely stem from data resolution differences. Sea ice compositions, recorded at segments of 10 cm for bulk salinity, $\delta^{18}O$, and [Nd], and 20 cm for $\varepsilon_{Nd}$, represent averages over approximately 14 to 28 days of ice growth and drift. In contrast, surface seawater samples were collected according to scheduling availability. Adjusting for permeability effects (Supplementary Text 3) could improve salinity alignment. However, better seawater–sea ice correspondence for $\varepsilon_{Nd}$ may be less visible due to limited vertical resolution in sea ice, sparse surface seawater data, and higher measurement uncertainties. Nonetheless, $\varepsilon_{Nd}$ signatures in sea ice correlate directly with surface seawater composition, as $\varepsilon_{Nd}$ remains unaffected by brine rejection or other sea ice processes. Intervals i) and iii) exhibit the least radiogenic $\varepsilon_{Nd}$ signatures, as low as $-10.8 \pm 0.5$, while intervals ii) and iv) show more radiogenic values of up to $-9.3 \pm 0.5$, reflecting variable mixing between Lena and Yenisei/Ob waters with AAW. The seawater $\varepsilon_{Nd}$ signature of $-10.3 \pm 0.1$ at 108 cm ice core depth (surface sample from station PS122/2_22-71) can be extrapolated to an approximate depth of 72 cm, corresponding to the sea ice thickness on December 12, 2019, when the low-salinity waters (S < 32) were first observed in winter (Fig. 3a). Given the high probability of these waters originating from a uniform shelf reservoir, their composition likely remained constant until February 10, 2020, when surface salinities rose above 32 (corresponding to about 114 cm ice core depth, Fig. 5a, c). The agreement within error between the extrapolated seawater $\varepsilon_{Nd}$ value and the sea ice signature establishes the link between the sea ice and surface water signatures for interval iii), along with the alignment across other intervals reinforcing that distinct sea ice intervals defined by

salinity and tracer distributions reflect variations in Siberian river water content and composition at the ice–ocean interface during sea ice drift.

To reconstruct the history of parental seawater through the first-year ice profile, we conducted a quantitative analysis of water mass endmember contributions to the sea ice ('Methods'). Intervals i) and iii) show nearly equal contributions from Lena and Yenisei/Ob waters, while intervals ii) and iv) are primarily dominated by Yenisei/Ob waters (Fig. 6a). The total river water volume in the ice ($f_{Lena} + f_{YenOb}$) aligns closely with the $f_{RIV}$ calculated from $\delta^{18}O$ and salinity for surface seawater, confirming the reliability of our methodology and the consistent incorporation of Siberian matter by the sea ice. Moreover, the dominance of Yenisei/Ob water over Lena water observed in the surface layer is also reflected in the sea ice, resulting in a combined ice–ocean compositional field within $f_{Lena}$–$f_{YenOb}$ space (Fig. 6b). Since seawater samples were not collected until after October 2019, before the MOSAiC ice floe had reached a thickness of about 40 cm, direct comparison between surface seawater and sea ice signals is only feasible for the younger ice intervals ii)–iv). However, the $\varepsilon_{Nd}$ signature of interval i) ($-10.8 \pm 0.5$) is, within uncertainty, identical to that of surface seawater collected near the North Pole in September 2020 ($-10.7 \pm 0.2$, station PS122_5_62-38). This similarity, also evident in $\delta^{18}O$ and salinity (Fig. 5a, b), suggests that the parental seawater for interval i) possessed characteristics akin to waters sampled near the North Pole. These waters, with elevated $f_{Lena}$ values, likely originated from the eastern Laptev Sea and passed through the region to its north, where first-year ice formation began. Their presence near the North Pole confirms the potential circulation of such waters in this area (Fig. 1). Snow infiltration can be eliminated as a major contributor to the tracer signatures in interval i), as the $\delta^{18}O$ ($-6.59$‰), $\varepsilon_{Nd}$ ($-8.0 \pm 0.2$), and distinct REE patterns (Supplementary Fig. S3) in our snow sample do not match the sea ice compositions. Other snow samples from the MOSAiC campaign had even lower $\delta^{18}O$ values, averaging $-16.3$‰[58]. While spatially variable snow infiltration associated with early winter thaw-freeze events could explain $\delta^{18}O$ values as low as $-6$‰ in the upper 20 cm of some first-year ice cores from the $\delta^{18}O$ timeseries[58], the $\delta^{18}O$ values of interval i) ($-1$‰) and the opposing

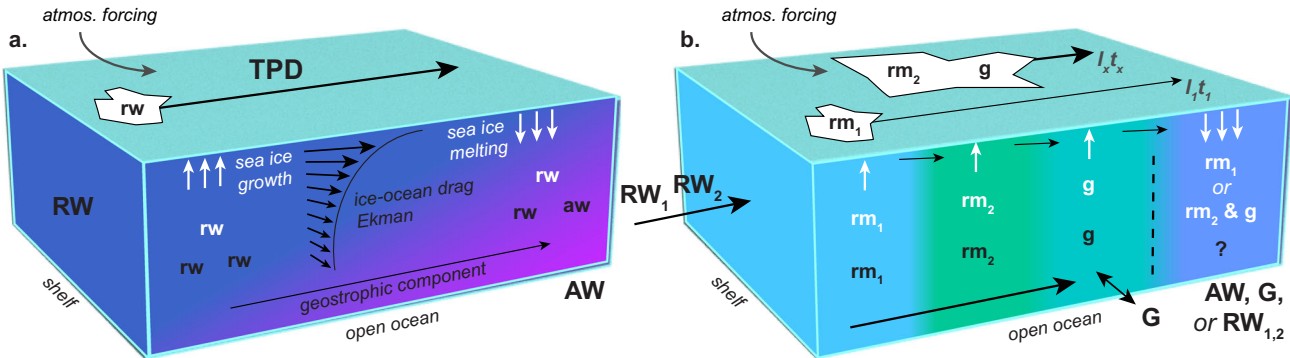

**Fig. 7 | Schematic comparison illustrating contrasting perspectives on TPD ice–ocean connectivity and ocean surface variability as primary drivers of matter redistribution. a** Simplified present perspective: This often implicitly adopted view stems from a limited understanding based on isolated assessments of ice and ocean reservoirs and their governing mechanisms. It emphasizes direct transport of matter from the Siberian Shelf to the Fram Strait or the Canadian Arctic Archipelago. In this representation, the Transpolar Drift (TPD) is depicted as a largely continuous pathway where a relatively homogeneous mix of Siberian river water (*RW*) and associated matter (*rw*) flows across the Arctic Ocean, both in the water column and within sea ice. The matter gradually blends with Atlantic-derived water (*AW*) and its constituents (*aw*), following a straightforward route. This approach concentrates on the bulk transfer of Siberian matter, with minimal differentiation among source contributions along the path. **b** Emerging perspective: While the transport mechanisms depicted in the simplified view remain valid, this perspective from our study emphasizes the dynamic and intricate nature of non-linear matter transport along the TPD. It highlights the variability in surface water composition and complex ice–ocean interactions. In this view, distinct matter mixtures ($rm_1$, $rm_2$) associated with various Siberian river waters ($RW_1$, $RW_2$), and additional sources, such as Greenlandic meltwater constituents (*g*) from Greenland (*G*), actively shape the composition of both seawater and sea ice at different points ($l_1$, $l_x$) and times ($t_1$, $t_x$) along the drift, with a future decrease in the long-range $l_1t_1$ pathway. The ice-driven redistribution of matter in the water column is indicated by white letters. Although both scenarios account for variable sea ice drift, straight arrows are used to provide a steady reference amidst surface layer fluctuations. Surface stresses are illustrated in (**a**) by arrows representing the Ekman spiral, alongside geostrophic flow, which together create an offset between ice drift and surface water flow. In (**b**), this offset becomes more pronounced as liquid freshwater transport is more constrained by geostrophic forcing, leading to a stronger decoupling between ice-driven and ocean-driven matter transport (represented by a dashed line).

$\varepsilon_{Nd}$ shift indicate that our cores were unaffected by substantial snow melt. Spring melt can also be ruled out, as the melt season had not begun by April 8, 2020, when the sea ice was recovered.

While sea ice values align closely with salinity and isotopic tracer signatures in surface water, differences are evident in the distribution of [Nd] and the PAAS-normalized HREE/LREE ratio (Fig. 5d, e). The [Nd] in sea ice constitutes approximately 5–7% of the corresponding seawater [Nd] in intervals i) and iii), and 9–10% in intervals ii) and iv), indicating variable REE incorporation in sea ice brine depending on the composition of the parental seawater. These variations seem linked to the presence (intervals i) and iii)) or absence (intervals ii) and iv)) of Lena water, highlighting the influence of the distinct river chemistries[59], which likely involve variations in REE complexation by inorganic and organic ligands[53]. The speciation of LREEs and HREEs in seawater and sea ice brine remains poorly constrained but is suspected to cause stronger HREE rejection during sea ice formation, resulting in a lower HREE/LREE ratio in sea ice compared to seawater[29,30] (Fig. 5e, Supplementary Fig. S3). However, despite this preferential rejection, the differences in HREE/LREE ratios between the parental source waters persist. The lower ratio in interval i) relative to other intervals aligns with a lower ratio in the surface water (Fig. 5e), likely due to a stronger contribution of Lena water (see previous section). These findings emphasize the importance of parental seawater composition in shaping the often highly heterogeneous distribution of substances in sea ice, such as microplastics[5], alongside the unique physicochemical processes that influence their incorporation and retention[60].

## Implications for dispersal of Siberian matter

Our research highlights the critical role of ice and water dynamics in redistributing Siberian matter across and beyond the TPD. The hydrographic variability of the Siberian Shelf regulates the export of fresh water and associated matter into the TPD, where limited mixing preserves distinct riverine contributions. This leads to spatial and temporal variability in downstream matter transport, driven by a near-surface geostrophic current primarily confined to the Amundsen Basin but potentially extending southward to the North Atlantic via the East

Greenland Current. Atmospheric changes can modify this current's flow and trajectory[14], affecting dispersal. However, divergent sea ice drift along the TPD exerts a much broader influence, extending matter dispersal far beyond the freshwater-dominated region over an annual cycle. Rapid exchanges at the ice–water interface throughout winter, driven by shifts in mixed-layer composition, highlight the efficiency of this redistribution process. Over a 200-day ice growth period, tracer distributions in sea ice and surface waters showed strong alignment, observed at 10 cm segments in the ice—representing approximately 14 days of ice growth and drift. This alignment emphasizes the rapid integration of river water and associated matter into sea ice within similar or shorter timescales. The REE data from the MOSAiC expedition exemplify this efficiency, revealing the storage of REEs from multiple Siberian rivers in sea ice within the Amundsen Basin during winter 2019/2020 and their subsequent release into the Atlantic Water-dominated Nansen Basin north of Svalbard by late April 2020. While dissolved REEs are present in trace amounts and pose minimal ecological risk, their long-range transport via ice drift and surface circulation illustrates the critical role of these processes in redistributing matter. These mechanisms collectively govern the movement of various forms of matter, ranging from dissolved substances to suspended particles. Notably, even suspended riverine particles, despite their size and density, have been detected far from their original sources[61,62], showcasing the extensive reach and efficiency of these processes. However, variability in source inputs— particularly for anthropogenic pollutants—and differing behaviors among substances complicate the quantification and prediction of dispersion pathways based solely on observed distributions, such as those of REEs. Recognizing these challenges, our study highlights the importance of understanding ice drift and circulation dynamics to evaluate the TPD's role in matter redistribution more effectively.

The TPD has traditionally been viewed as a straightforward conduit for transporting matter—via water or ice—from Siberian coasts to the Fram Strait (Fig. 7a). This perception has largely been based on limited winter observations and a tendency to study sea ice[3–6] and ocean[7,8] reservoirs separately. However, our findings reveal a more intricate and dynamic transport network (Fig. 7b). Variations in the

velocity and direction of ice and ocean flows, combined with fluctuating freshwater inputs from the Siberian Shelf and seasonal ice–ocean interactions, redistribute matter far beyond its original terrestrial and marine sources. Sea ice formed along the TPD plays a pivotal role in this system, integrating contributions from multiple sources during extended drifts and significantly enhancing matter dispersal. Conversely, sea ice originating from the Siberian Shelf often carries matter from a single terrestrial source—dependent on whether contributions from that source were present in surface waters on the shelf during the autumn and winter ice formation periods. This evolving understanding expands the TPD's role to encompass not only Siberian inputs but also other terrestrial contributions. For instance, meltwater from Greenland has been detected in sea ice floes sampled across the Fram Strait, even when reconstructed drift trajectories indicate ice origin and transport far from Greenland's coast[29]. This suggests that Greenland-derived matter can be advected deep into the TPD's influence zone before being incorporated into the ice, further underscoring the critical role of the complex and interconnected ice–ocean pathways within the TPD system.

As Arctic sea ice transitions from multi-year ice to predominantly seasonal ice, observational methods and modeling strategies must adapt. Reduced sea ice extent on the Siberian Shelf and along the TPD will create more variable liquid freshwater pathways, increasingly governed by atmospheric conditions[18]. A more fragmented ice pack will also heighten the responsiveness of sea ice drift to atmospheric forcing[18,63], intensifying ice-driven dispersal of Siberian matter. With the decline of long-range transport of ice-rafted matter from the Siberian coast to the Fram Strait[35], seasonal ice-driven redistribution along the TPD will become increasingly important (Fig. 7b). Climate-driven increases in river runoff[64,65] and shifts in river water chemistry[10,66] will further modify the transport of matter through these evolving pathways. Improving our understanding of long-range ice–ocean connectivity and water column variability is therefore crucial for refining predictions of the transport of natural substances and anthropogenic pollutants within the rapidly changing Arctic system.

## Methods

### Sampling and pre-treatment

Seawater and sea ice samples were collected from the MOSAiC Central Observatory, which features sampling facilities positioned on the main sea ice floe. These included ocean sampling sites, the *RV Polarstern* vessel, and an ice hole located several hundred meters away, referred to as *Ocean City*[32]. Over five research legs from October 2019 to September 2020, a total of 41 large-volume seawater samples (10 L; hereafter referred to as *isotope samples*) were collected using a Conductivity-Temperature-Depth (CTD) rosette fitted with 24 OTE (*RV Polarstern* samples) and 12 (*Ocean City* samples) Niskin bottles. The corresponding CTD bottle data have been published by Tippenhauer et al.[67,68]. At each station, seawater samples were obtained from the surface (0 to 5 m) and depths of 50, 100, and 200 or 250 m. The samples were collected in acid-cleaned LDPE-cubitainers and transported to the clean room laboratory at GEOMAR (MK-Versuchsanlagen, class 100 hoods). In the laboratory, filtration through 0.45 Merck Millipore® cellulose acetate filters was carried out using a peristaltic pump. After filtration, samples were acidified to pH -2.2 with ultra-pure concentrated hydrochloric acid. Small aliquots (0.1 L) for $\delta^{18}O$ analysis were obtained before filtration and acidification but after homogenization and stored in glass bottles at 4 °C until analysis. A set of seawater samples for $\delta^{18}O$ analysis with a higher spatial resolution was collected separately in 0.2 L glass bottles from the Central Observatory using the same sampling facilities.

In addition to seawater, ten sea ice cores were drilled in proximity to each other with a Kovacs 9 cm diameter corer (Kovacs Enterprise, Roseburg, USA) on April 8, 2020, from the main coring site (MCS) for first-year ice and second-year ice (station PS122/3_35-80). These cores were promptly placed into plastic bags (LDPE tube films by Rische and

Herfurth) and stored at −20 °C along with a snow sample, collected in a clean LDPE bag prior to core extraction. Upon arrival at the clean room laboratory at GEOMAR, the ice cores underwent a rinsing process with deionized water (Milli-Q, 18.2 MΩcm) at a temperature of approximately 20 °C for around 15 seconds to remove contaminants from the core surface. Similar rinsing procedures are common practice in the preparation of glacial ice cores for tracer analysis[69] and serve to eliminate loose surface contaminants that may not be completely removed by physical scraping alone. The rinse does not affect the brine content of the ice core, as only enclosed brines remain post-sampling, which are not altered by brief exposure to deionized water. After the rinse, careful scraping was conducted using a custom-made titanium grade 1 chisel to remove the wet top surface and any remaining contaminants. The cores were then sectioned into 10 cm pieces using a custom-made titanium grade 1 handsaw on a clean plastic table lined with acid-cleaned PFA foil. The resulting segments from nine ice cores, corresponding to specific depths, were combined in acid-cleaned LDPE buckets with sealed plastic lids for melting. The combination of core segments of equal depth was implemented to increase the sample volume needed for Nd isotope analysis while also mitigating the impact of sea ice heterogeneity and brine spillage during sampling (Supplementary Text 3). Although this approach cannot eliminate the effects of brine spillage during field core retrieval—a process known for its potential to cause significant brine loss—it reduces them by increasing the proportion of trapped brine relative to spilled brine. This is evidenced by the consistency of our sea ice tracer profiles derived from the combined cores. To further evaluate the effects of heterogeneity and brine loss, samples from one core were processed separately from the other nine. These individual 10 cm samples were melted in acid-cleaned LDPE bottles, which were cut open using an acid-cleaned ceramic knife to insert the ice core samples, and subsequently sealed with tight-fitting acid-cleaned LDPE covers. No leakage of brine was observed during the entire sample processing in the laboratory. Following the complete melting of the final piece of ice after approximately 12 h, the meltwater was filtrated through 0.45 μm Merck Millipore® cellulose acetate filters and directly transferred into acid-cleaned LDPE cubitainers (for pooled samples from nine ice cores) and acid-cleaned LDPE bottles (for individual samples from one ice core). After homogenization and sub-sampling for salinity and $\delta^{18}O$ analysis, the filtered samples were acidified to pH ≈ 2.2 using ultra-pure concentrated hydrochloric acid. A minimum equilibration period of 48 h was allowed before another aliquot was separated into an acid-cleaned LDPE bottle for REE and Nd concentration analyses. The snow sample underwent a similar treatment, excluding the deionized water rinse and sectioning steps.

### Neodymium isotopes of seawater, sea ice, and snow

The preparation of samples for $\varepsilon_{Nd}$ and REE analyses at the GEOMAR laboratory adhered strictly to the established GEOTRACES protocols and underwent validation through participation in the international GEOTRACES intercalibration study[70]. The pre-concentration of Nd involved Fe co-precipitation: A trace metal clean $FeCl_3$ solution (-200 mg Fe ml⁻¹) was added to each seawater, sea ice, and snow sample. Following a 24 h equilibration period, trace metal clean ammonia solution (25%, Merck Suprapur®) was added to raise the pH to 7.5–8.0. After 48 h, the precipitated trace elements, together with FeOOH, settled to the cubitainers bottom, and the supernatant water was suctioned off. The precipitates were centrifuged and rinsed three times with deionized water (MilliQ, 18.2 MΩcm) to eliminate major ions. They were then transferred to PFA vials with 6 M HCl and evaporated to dryness. To eliminate organic components, the samples underwent treatment with aqua regia at 120 °C for 24 h. After evaporation to dryness, the samples were dissolved in 6 M HCl. They were then washed twice with pre-cleaned diethyl ether to remove approximately 99% of the Fe, which forms an ethereal complex that can be

separated from the acidic phase containing the trace metals. Following separation, the samples were dried down again and dissolved in 1 M HCl for column chemistry. REEs were separated from matrix elements through cation exchange chromatography (BIORAD®, AG50W-X8 resin, 200–400 μm mesh-size, 1.4 ml resin bed) with a modified separation scheme based on Laukert et al.[22]. Further purification of Nd from other REEs for isotope measurements was accomplished using Eichrom® LN-Spec resin (2 ml, 50–100 μm) following established procedures[22]. To eliminate residual traces of the resin and organic compounds, the samples underwent a final treatment with concentrated quartz-distilled $HNO_3$ before $\varepsilon_{Nd}$ measurements.

The $^{143}Nd/^{144}Nd$ ratios were measured on a Neptune Plus MC-ICP-MS at GEOMAR for 40 cycles at 4 s and concentrations between 10 and 30 ppb Nd for seawater, 2 to 4 ppb Nd for sea ice ($10^{12}$ Ω resistors assigned to masses 143 and 146), and 7 ppb for the snow sample in 380 μL solution. Since all sea ice samples prepared for the $\varepsilon_{Nd}$ analysis had too low concentrations (<0.7 ng) despite the combination of 10 cm depth segments from nine ice cores, samples from two (0–100 cm and 130–150 cm) or even three (100–13 cm) depth segments had to be combined to achieve concentrations of at least 1 ng and external uncertainties below 0.5 $\varepsilon_{Nd}$ units (2 s.d., see below). Instrumental mass bias was double-corrected with $^{146}Nd/^{144}Nd = 0.7219$ and $^{142}Nd/^{144}Nd = 1.141876$, with the condition that the $^{142}Ce$ beam intensity was sufficiently low[71]. This was monitored by ensuring a raw measured $^{140}Ce/^{144}Nd <1$ (i.e. Ce/Nd <0.3) for each sample and standard solution. Isobaric interferences between $^{144}Sm$ and $^{144}Nd$ were corrected by measuring the abundance of the interference-free isotope $^{147}Sm$ and by calculating the potential $^{144}Sm$ contribution on mass 144 from the natural abundance of Sm. The $^{143}Nd/^{144}Nd$ ratios of all samples were normalized to the accepted jNdi-1 standard value of 0.512115[72]. The external reproducibility of the Nd isotope measurements as estimated by repeated measurements of an in-house laboratory standard with concentrations matching those of the measured samples ranged between 0.15 and 0.3 $\varepsilon_{Nd}$ units (2 s.d.) for seawater and 0.37 and 0.5 for sea ice. The external reproducibility of the snow sample was 0.23 $\varepsilon_{Nd}$ units (2.s.d.). Blanks ($n = 6$) were processed identically to the samples and had <5 pg Nd, which corresponds to <0.7% (sea ice/snow) and <0.4% (seawater) of the lowest Nd concentration used for the isotope measurements. Blank corrections were therefore not applied. The $\varepsilon_{Nd}$ is defined by Eq. (1) as follows:

$$\left[ \frac{(^{143}Nd/^{144}Nd)_{sample}}{(^{143}Nd/^{144}Nd)_{CHUR}} - 1 \right] \times 10^4 = \varepsilon_{Nd} \qquad (1)$$

where the $^{143}Nd/^{144}Nd$ of CHUR ('CHondritic Uniform Reservoir') is 0.512638[73].

### Neodymium concentrations of seawater (isotope dilution method)

0.5 L aliquots of the acidified and filtered seawater samples were spiked with a pre-weighed $^{150}Nd$ spike, preconcentrated using Fe co-precipitation, and purified by column chromatography using the same method as for the $\varepsilon_{Nd}$ measurements, with the difference that only cation exchange chromatography was applied. The isotope dilution measurement of the Nd concentration based on $^{150}Nd/^{144}Nd$ ratio was carried out on Nu Plasma II MC-ICP-MS at GEOMAR. External reproducibility (2 s.d.) was better than 0.8% according to repeated treatment and measurement of the same sample ($n = 3$). Blanks ($n = 2$) exhibited Nd concentrations <4 pg, obviating the need for correction.

### Rare earth element concentrations of seawater, sea ice, and snow

REEs were pre-concentrated at GEOMAR offline using a SeaFAST system (model M5 from Elemental Scientific), following a method refined from Hathorne et al.[74]. Using this updated method, 12 ml (seawater) or

24 ml (sea ice segments, snow) of the acidified sample was introduced to the column via a fifth syringe pump. After matrix removal, the REEs were eluted twice with 200 μL of 1.5 M $HNO_3$. Before pre-concentration, each blank, reference material, and sample (pH ~2) underwent spiking with 12 μL of thulium solution (10 ng g$^{-1}$) for yield monitoring, typically achieving for seawater 97 ± 4.6% yields (± 1 s.d., $n = 141$) and for sea ice 92 ± 3.7% yields (± 1 s.d., $n = 92$). Before analysis on a Thermo Element XR ICP-MS coupled with a CETAC "Aridus" desolvating nebulizer, all samples were diluted with 200 μL of 0.1% $HNO_3$ containing 10 ng g$^{-1}$ Re as an internal standard during measurement and to account for any sample evaporation post pre-concentration. The desolvating nebulizer enhanced sensitivity and reduced oxide formation, monitored with Ba, Ce, Pr + Nd, and Sm + Eu + Gd + Tb element solutions at the beginning of each analytical session. Oxide formation remained generally <0.01 ($n = 3$) for Ba, <0.05 ($n = 3$) for Ce, <0.04 ($n = 3$) for Pr + Nd, and <0.04 ($n = 3$) for the MREE. GEOTRACES inter-calibration sample BATS 15 m and in-house reference seawater underwent the same pre-concentration as seawater, sea ice and snow samples to monitor external reproducibility and accuracy. Additionally, certified natural river water reference material SLRS-6[75] was diluted ~500 times as a standard for sea ice and snow measurements at similarly low REE concentrations, and an in-house reference seawater served as an additional standard for seawater samples. The measured REE values of BATS 15 m and SLRS-6 align well with literature values. External reproducibility for seawater data based on BATS 15 m ($n = 17$) was better than ~6% for the LREEs (from La up to and including Eu), and better than ~5% for MREEs (from Gd up to and including Dy) and HREEs (from Ho up to and including Lu). For sea ice and snow, external reproducibility based on diluted SLRS-6 ($n = 6$) was better than ~11%, ~13%, and ~30% for LREEs, MREEs, and HREEs, respectively. Procedural laboratory blanks for seawater ($n = 4$) had REE concentrations, on average, corresponding to ~3% of samples analyzed, except for Eu, which averaged ~8%. For sea ice and snow, the blanks ($n = 4$) had concentrations on average corresponding to ~9% of the samples, except for Eu ( ~19%).

### Stable oxygen isotopes of seawater, sea ice, and snow; sea ice bulk salinity

Oxygen isotope compositions in the seawater aliquots obtained from the *isotope samples* and in sea ice (pooled samples from nine ice cores and individual samples from one ice core) as well as the snow sample were analyzed at the Stable Isotope Laboratory of the College of Earth, Ocean, and Atmospheric Sciences at Oregon State University (Corvallis, USA) by applying the $CO_2$-water isotope equilibration technique[76] on at least 2 sub-samples on a Finnigan gas bench II unit coupled to a Finnigan DeltaPlusXL. Oxygen isotope compositions in the more extensive set of seawater samples[41] were mainly analyzed at the Leibniz Laboratory (Kiel, Germany), applying the same technique. The external reproducibility for the $\delta^{18}O$ measurements was ± 0.05‰ or better for sea ice, snow and aliquots from the *isotope samples* and ± 0.04‰ or better for the more extensive seawater dataset generated at the Leibniz Laboratory. A small subset of the more extensive seawater $\delta^{18}O$ dataset was analyzed at the AWI with external reproducibility of ± 0.07‰. The measured $^{18}O/^{16}O$ ratio is provided as a deviation from V-SMOW ('Vienna Standard Mean Ocean Water') in the δ-notation following Craig[77] with Eq. (2):

$$\left[ \frac{(^{18}O/^{16}O)_{sample}}{(^{18}O/^{16}O)_{V-SMOW}} - 1 \right] \times 10^3 = \delta^{18}O \qquad (2)$$

Where $(^{18}O/^{16}O)_{sample}$ and $(^{18}O/^{16}O)_{V-SMOW}$ are the $^{18}O/^{16}O$ ratios of the sample and the V-SMOW, respectively.

The bulk salinity of the sea ice and snow was determined with an AutoSal 8400 A salinometer at GEOMAR with a precision of ± 0.003 and an accuracy of better than ± 0.005.

## Water mass analysis

We assume that the waters encountered during the MOSAiC drift campaign are different mixtures of Atlantic water (AW), Lena river water (Lena), and a mix of Yenisei and Ob river water in discharge-weighted proportions (YenOb). In addition, we account for sea ice meltwater (SIM) or brine (negative SIM) contributions to the water column. The rationale for the selection of these endmembers and the potential contribution of other endmembers, such as other Siberian or Canadian rivers, glacial meltwater and Pacific water, is discussed in Supplementary Text 2. To determine the proportions of the selected endmembers, we employ mass balance calculations solved through matrix inversion, combining salinity, $\delta^{18}O$ and $\varepsilon_{Nd}$ signatures, as outlined in Eqs. (3) to (6):

$$f_{AW} + f_{Lena} + f_{YenOb} + f_{SIM} = 1 \tag{3}$$

$$f_{AW} \times S_{AW} + f_{Lena} \times S_{Lena} + f_{YenOb} \times S_{YenOb} + f_{SIM} \times S_{SIM} = S_{sample} \tag{4}$$

$$f_{AW} \times \delta^{18}O_{AW} + f_{Lena} \times \delta^{18}O_{Lena} + f_{YenOb} \times \delta^{18}O_{YenOb} + f_{SIM} \times \delta^{18}O_{SIM} = \delta^{18}O_{sample} \tag{5}$$

$$\frac{f_{AW} \times [Nd]_{AW} \times (^{143}Nd/^{144}Nd)_{AW} + f_{Lena} \times [Nd]_{Lena} \times (^{143}Nd/^{144}Nd)_{Lena} + f_{YenOb} \times [Nd]_{YenOb} \times (^{143}Nd/^{144}Nd)_{YenOb} + f_{SIM} \times [Nd]_{SIM} \times (^{143}Nd/^{144}Nd)_{SIM}}{f_{AW} \times [Nd]_{AW} + f_{Lena} \times [Nd]_{Lena} + f_{YenOb} \times [Nd]_{YenOb} + f_{SIM} \times [Nd]_{SIM}} = \left(^{143}Nd/^{144}Nd\right)_{sample} \tag{6}$$

Here, $f_{endmember}$ represents the fraction of the respective end-member in the sample, while $Tracer_{endmember}$ denotes their respective tracer values. The measured values for each sample are represented by $Tracer_{sample}$. Our four-component analysis differs from the Nd-based approach of Paffrath et al.[25], which relied on an Nd concentration balance based on sample [Nd] to distinguish between four water masses (AW, Pacific water, Lena and Yenisei/Ob). This method introduced sensitivity issues due to small variations in riverine [Nd], which significantly affected the calculated fractions, making attempts to quantitatively differentiate these water masses unreliable. Instead, we only use the endmember [Nd] to account for the average-weighted isotopic composition in Eq. (6), while excluding sample [Nd] from our analysis. This is possible because we were able to exclude Pacific water in our study region (see Supplementary Text 2.4). This improves the robustness of our method and allows for a clearer assessment of non-conservative processes influencing $\varepsilon_{Nd}$ and [Nd] beyond estuarine REE removal (see Supplementary Text 1). We have also refined the endmember definitions established by Paffrath et al.[25], adjusting for scatter caused by natural (seasonal) variability, estuarine Nd removal, and incorporating analytical uncertainties where direct data on variability were unavailable (Supplementary Table S1). To account for these uncertainties, we applied a Monte Carlo simulation, generating 100,000 sampled values within the defined parameter space for the system of Eqs. (3–6). This provides a more statistically sound evaluation of water mass contributions and tracer behavior. The errors in Figs. 4–6 reflect the uncertainties derived from the Monte Carlo simulation. Slightly negative $f_{Lena}$ values (<1%) cannot be fully attributed to these uncertainties and are likely linked to an overestimation of $[Nd]_{AW}$, even after considering Nd removal processes during the transit of AW through the relatively productive Barents Sea (Supplementary Text 2.1). Adjusting $[Nd]_{AW}$ to values below 11 pmol kg$^{-1}$ would slightly alter the relative proportions of the water mass fractions but would not impact the overall conclusions of this study.

The total river water fraction ($f_{RIV}$, here: representing the combined contribution of the Lena and YenOb endmembers) can also be determined using salinity and $\delta^{18}O$ alone[8]. This three-component analysis (i.e. RIV, AW, SIM) is accomplished by adjusting Eqs. (3–5) to account for $f_{Lena} + f_{YenOb} = f_{RIV}$, and by combining the parameter values for both rivers. This enables the determination of $f_{RIV}$, for example, using the more extensive $\delta^{18}O$ datasets[41,42], for which no $\varepsilon_{Nd}$ data are available, offering insights into river water distribution (e.g., Fig. 3a) that cannot be obtained from the 4-component analysis above due to the lower resolution of $\varepsilon_{Nd}$ samples. The $f_{RIV}$ derived from this approach are within error identical to the combined fractions of Lena and YenOb from the $\varepsilon_{Nd}$-based analysis, as $\varepsilon_{Nd}$ primarily serves to differentiate between the two river endmembers. Likewise, the $f_{SIM}$ values from both analyses agree within error. We assume negligible in situ precipitation in the central Arctic Ocean compared to river discharge volume[7], which is supported by the strong correlation between $f_{RIV}$ and dissolved [Nd] (Fig. 4a).

## Alignment of sea ice and ocean surface signatures; endmember contributions to sea ice

Accurate assignment of surface seawater signatures to the sea ice profile requires an age model for the sea ice, which we determine by a linear regression that correlates ice thickness with measurement date ($R^2 = 0.97$, $p$-value <0.05, $n = 56$). This correlation is based on data from regular sampling at the Central Observatory between November 7, 2019, and May 9, 2020, during which a consistent linear growth rate of ca. 0.3 mm h$^{-1}$ was observed for first-year ice[39]. During the initial growth phase in October and early November 2019, the sea ice either formed very slowly or did not form at all, maintaining a consistent first-year ice thickness of $40 \pm 2.2$ cm ($\pm 2$ s.d., $n = 14$), based on measurements at the MCS during this period. This lack of significant growth is not captured by the age model, which would otherwise predict unrealistic thicknesses below 40 cm for this period. To account for this, we assigned the average thickness of 40 cm for the first seawater sampling on October 4, 2019, and a thickness of 42 cm for the seawater sampling on October 24, 2019, measured at the MCS site on the same day. As we lack surface seawater data for the upper 40 cm of the sea ice profile, variations in ice growth rate within this depth are considered irrelevant for our study. The accuracy of the ice core depth is assessed by an analysis of multiple depth measurements from different ice cores taken on the same day[39], resulting in an estimated uncertainty of 10 cm (2 s.d.), which is indicated in all relevant figures and factored into our interpretation of the data.

The observed consistent deviation of about 2‰ in $\delta^{18}O$ between the values for seawater at the surface and sea ice within the first-year ice core depth of $40-150$ cm can be explained by $\delta^{18}O$ fractionation during sea ice growth[8]. The equilibrium fraction value for $\delta^{18}O$ in sea ice exhibits variability ranging from <0.1‰ for fast-growing (>10 mm h$^{-1}$) frazil ice to >2.5‰ for slow-growing (<0.01 mm h$^{-1}$) column ice[78]. Considering the linear growth rate of about 0.3 mm h$^{-1}$ for the first-year ice, the estimated equilibrium fractionation value of −2‰ agrees well with other field observations when the ice growth rate is considered[79].

The partition coefficient for salinity between surface seawater and sea ice is determined to be $0.13 \pm 0.1$ (2 s.d., $n = 28$) and is calculated for the ice core depth at which steady-state salinity is reached (i.e. excluding the lowermost 10 cm), based on a regression line derived from the average salinities of seawater and sea ice at depths of 40–70 cm and 90–100 cm within the ice core. The salinity representing the surface seawater salinity during sea ice formation reconstructed from the regression line lies between 30.56 and 32.91, which agrees well with the salinity of the surface seawater sampled during the

corresponding drift interval. Using this salinity, we calculate the expected salinity without the effects of sea ice formation or melting following Rosén et al.[80], and by adjusting the $^{18}O/^{16}O$ ratio of the sea ice sample by −2‰ for equilibrium fractionation. The corrected salinity is then used, along with $\varepsilon_{Nd}$ and $\delta^{18}O$, to determine the contribution of the three endmembers to sea ice, applying Eqs. (3–6). Here, $(^{143}Nd/^{144}Nd)_{sample}$ and $\delta^{18}O_{sample}$ represent the measured $^{143}Nd/^{144}Nd$ ratio in the sea ice and the fractionation-corrected $\delta^{18}O$. The calculated $f_{SIM}$, ideally expected to be zero, is effectively zero within error, validating the approach. Here, too, a Monte Carlo simulation with 100,000 sampled values was used to determine the fraction errors associated with endmember variability and measurement uncertainty.

The differences in sea ice sampling resolution between salinity, $\delta^{18}O$ and [Nd] (10 cm) compared to $\varepsilon_{Nd}$ signatures (20 cm) lead to deviations in their trends at the 30–40 cm depth horizon in the ice profile. While the $\delta^{18}O$, [Nd] and HREE/LREE values at this depth align more closely with other values of interval ii), the $\varepsilon_{Nd}$ signature is more consistent with the uppermost signature of interval i) (Fig. 5). From this comparison, we infer that the $\varepsilon_{Nd}$ signature of the 40–60 cm ice core depth better represents the 30–40 cm depth horizon in the water mass analysis than the signature reported for the 20–40 cm ice core depth. Therefore, we adjust the $\varepsilon_{Nd}$ signature for the 30–40 cm depth horizon to match that of the 40–60 cm ice core depth in our analysis. The $f_{Lena}$ and $f_{YenOb}$ values derived from this sea ice sample, as shown in Fig. 6, are based on this corrected $\varepsilon_{Nd}$ signature. After this correction, the sample falls within the data envelope in $f_{Lena}$:$f_{YenOb}$ space in Fig. 6b, supporting the validity of our approach. The excellent agreement between the calculated $f_{RIV}$ in sea ice and seawater (Fig. 6a) confirms the suitability of our water component analysis in sea ice.

## Data availability
The tracer data generated in this study have been deposited in the PANGAEA database (isotope samples: https://doi.org/10.1594/PANGAEA.966223, https://doi.org/10.1594/PANGAEA.966225; more extensive $\delta^{18}O$ data sets: https://doi.org/10.1594/PANGAEA.966184, https://doi.org/10.1594/PANGAEA.948291, https://doi.org/10.1594/PANGAEA.958464). The CTD bottle data corresponding to the tracer samples and the continuous salinity are also available in the PANGAEA database (CTD bottle data from *RV Polarstern*: https://doi.org/10.1594/PANGAEA.959965; CTD bottle data from Ocean City: https://doi.org/10.1594/PANGAEA.959966; continuous CTD salinity from *RV Polarstern*: https://doi.org/10.1594/PANGAEA.930023; https://doi.org/10.1594/PANGAEA.930024; https://doi.org/10.1594/PANGAEA.930026; https://doi.org/10.1594/PANGAEA.930027; https://doi.org/10.1594/PANGAEA.930028).

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

## Acknowledgements

This research was conducted as part of the international Multidisciplinary drifting Observatory for the Study of Arctic Climate (MOSAiC2019–2020), utilizing data generated during the expedition of the Research Vessel Polarstern (AWI_PS122_00) in 2019–2020[81]. The authors express gratitude to all individuals involved in the MOSAiC expedition. Appreciation is also extended to Jutta Heinze and Sieglinde Kolbrink for their laboratory support and Marcus Gutjahr and Christopher Siebert for their analytical guidance. Martin Frank is gratefully acknowledged for providing financial support, covering sample transport and laboratory expenses. G.L. received funding from the Ocean Frontier Institute, supported by a Canada First Research Excellence Fund award (grant no. 39291), and from the European Union's Horizon 2020 research and innovation program under a Marie Skłodowska-Curie Postdoctoral Global Fellowship award (grant no. 101023769). Additional financial support was provided for D.B. by the German Science Foundation (grant no. BA 1689/4-1) and for A.D. from the U.S. National Science Foundation (grant no. 1821900). S.T. and M.V. were partly supported by the Changing Arctic Ocean program (grant nos. NE/R012865/1, NE/R012865/2, #03V01461 and #03F0804A), jointly funded by the United Kingdom Research and Innovation Natural Environment Research Council and the Bundesministerium für Bildung und Forschung. K.H. also acknowledges funding from the United Kingdom Research and Innovation Natural Environment Research Council through the Changing Arctic Ocean program (grant nos. NE/P005942/1, NE/P006108/1, NE/P006493/1) and the BIOPOLE project (grant no. NE/W004933/1). P.S.P acknowledges support from the Swedish Research Council Formas and the Swedish Polar Research Secretariat. T.J.H. acknowledges support from the U.S. National Science Foundation and the Woods Hole Oceanographic Institution's Ocean and Climate Innovation Accelerator program. S.K. and M.K. acknowledge continuous funding from the Natural Sciences and Engineering Research Council of Canada.

## Author contributions

G.L. designed and coordinated the study and conducted the analytical work on the *isotope samples*. D.B., E.D., P.S.P., A.D., and D.N. conducted the sampling. D.B. designed and coordinated the oxygen isotope part of the study and, together with H.M., M.M., and N.A., generated the extensive oxygen isotope data set with a higher spatial resolution. E.H. guided the analysis of the REEs. G.L. wrote the manuscript. D.B., B.R., T.K., E.D., M.K., E.H. M.V., S.T., N.A., H.M., M.M., A.D., P.S.P., D.N., T.J.H., K.H., and S.S.K contributed to the final version.

## Competing interests

The authors declare no competing interests.
