## [Transparent Peer Review file · Nature Communications]

Dynamic Ice–Ocean Pathways along the Transpolar Drift Amplify the Dispersal of Siberian Matter

Corresponding Author: Dr Georgi Laukert

Version 0:

Reviewer comments:

Reviewer #1

(Remarks to the Author)

Review of "Arctic Transpolar Drift Amplifies Siberian Matter Dispersal through Decoupled Sea Ice and Ocean Surface Pathways" by Laukert et al., submitted to Nature Communications

In this ms., the authors present a suite of novel neodymium and oxygen isotope, rare earth element and salinity data from a unique expedition sampling throughout a winter season in the Arctic ice. Their data are interpreted in terms of input of freshwater from the Siberian rivers to the Transpolar Drift (TPD), an important surface current that crosses the Arctic Ocean, into the highest North Atlantic. Their conclusions are fine, but rather ambiguous – statements that tend to amount to “there is complexity,” but not exactly what this means.

I fear that much of the perceived impact of this paper lies in the unique nature of the sampling campaign, which is an impressive feat. But the data generated and conclusions reached are almost identical to their prior publication in 2021 (Paffrath et al., Nature Scientific Reports). I was hoping that this more recent ms. would have had further, more definitive conclusions on the nature of distribution of riverine sourced tracers, and how they might impact the North Atlantic. Particularly because these data are interesting and I think that they should be published. However, as is, this ms. reads the same as their last (GEOTRACES) work. – perhaps a longer-format journal would be more appropriate for more detailed and deeper discussion of both these data sets?

Regardless of decision, I might offer the authors the following concerns to consider:

1. River water input is highly variable over the year, particularly in the Arctic. The winter has near-negligible discharge, compared to massive spring flows. There is no mention of this in the ms., but I would imagine it is important. Similarly, what is the residence time of river water in the shelf area, or in the TPD?
2. The systematics of eNd are oversimplified. This is not an open-ocean setting, and the residence time of Nd is not the same as deep, open ocean (it cannot be). The “quasi-conservative” nature of eNd in this setting cannot be assumed.
3. I would not be shocked to see 75% Pacific water at these depths. Saying it is “improbable” seems suspect. Pacific water penetrates very far into the Arctic, particularly in the winter when river flows are very low. Again, see point 1.
4. I do not trust any of the [Nd] arguments. There is nothing conservative about [Nd] to suggest it can be used in these ways – particularly when the “mixing” plot already assumes a 75% loss in the measured riverine end member (i.e., most of the Nd is lost in the estuary, but how consistently?). That they found a linear mixing curve is interesting, to be sure (dependent on some assumptions). But it makes little sense with respect to the isotopes; how can the concentration continue to increase with apparent equal sensitivity/change in the isotopic fingerprint? The downstream isotopes should not change as readily, unless there is an effective scavenging, which calls into question the [Nd] patterns.
5. I want to see the Pacific treated quantitatively in their ms., and shown to be invalid, in figures and calculations. They did this in their 2021 publication, but did not here. In neither publication is a $1/[Nd]$ vs. eNd plot shown, which should, if conservative, show clear mixing patterns.
6. The boundary exchange problem for eNd must be addressed. Boundary exchange has been shown to not impact [Nd] from original observations. The authors dismiss this citing lack of evidence in overlying water, but this is weak, and was indeed weak in the original boundary exchange publications.
7. Brine spillage – I appreciate the difficulties of sample collection, and this should not be held against the authors, but should be treated as honestly as possible. The fact that brine has far greater [Nd] concentrations than non-brine water makes a linear mixing plot even more confusing. This should be treated with greater discussion.
8. L68-69 needs references? This seems like a bold statement, given Ekman transport is one of the original “oceanographic”

theories, based on years of observation and is a cornerstone of physical oceanography. (The story of Nansen is the oceanography equivalent to Newton's apple...). Throughout the ms., bold statements confronting established physical oceanographic principles should be considered deeper.

In summary, I feel that there are a few critical points that should be addressed regardless of where this ms. is published. I do hope these data are published, and it would be great if published with Nature Communications - but I think this would take a significant re-write.

Reviewer #2

(Remarks to the Author)
Comments to the authors:

Laukert et al. present a well-written manuscript presenting geochemical provenance tracer data collected from the MOSAiC campaign in the Eurasian Arctic Ocean. This exciting dataset is potentially significant because it fills the knowledge gap of sea ice and ocean surface pathways in the polar winter. The manuscript is clear, thorough, and nearly free of typos. The authors have done a good job combining multiple chemical tracers and disentangling complex endmembers. However, I have some substantive comments, as listed below. One of my main concerns is that the authors may have mistakenly omitted the sea ice melt fraction in their mass balance Equation 3, which could generate quite different results from the current version, at least during part of the cruise. Overall, I recommend publication with major revisions.

%-----

Major interpretation points

This reviewer has some major questions about Equations 3 to 5. First, the authors should add f_{ice} in Equation 3 if I correctly understand Rosen et al. (2015)'s Equation 3. Without f_{ice} , the sum of f_{AW} , f_{Lena} , and f_{YenOb} wouldn't be 1. Secondly, the authors correct for salt fluxes related to sea ice formation and melting for salinity in Equation 4, but no such correction exists for Equation 5. The authors could argue that neodymium isotopes remain unfractionated during sea ice processes, but sea ice processes do contribute to Nd concentrations, as seen in Figure 5d. This would change the denominator of Equation 5. Thirdly, if my previous two points are correct, I am curious how these fractions would change after taking into account the fraction of sea ice melt in Equations 3 and 5. I don't think the general story would change, but there could be some changes to some interpretations, at least during part of the cruise.

%-----

Interpretation points, by line #

Lines 31–32: The decoupling between sea ice and ocean surface transport doesn't seem to only occur in the winter.

Lines 50–51: The Fram Strait is not the only outflow of the TPD. What about the outflow into the Canadian Arctic Archipelago (CAA)? For example, the authors can check out their own schematic Figure S3 or Carmack et al. (2016) (10.1002/2015JG003140).

Lines 206–208: The authors should consider plotting d_{18O} and Nd section plots in the supplemental. It's not immediately obvious if the Pacific water has some influence on the subsurface water mass, especially for those close to the Makarov Basin, which can be identified by Paffrath et al. (2021). Additionally, how would the Greenland glacier runoff influence ($\epsilon_{Nd} < -17$; Figure S3) the end-member mixing model close to the Fram Strait?

Lines 335–337: It's hard to reach this conclusion with such a limited number of Nd or REE measurements as compared to salinity and d_{18O} . I am also not sure why the authors have different conclusions between the pattern of ϵ_{Nd} (e.g., Lines 292–295) and Nd or HREE/LREE. All of those parameters only have 5 points in Figure 5. Are there any statistical reasons?

Line 344: Why does this have implications for particles? All parameters discussed in the manuscript are dissolved. Particles tend to sink, so they cannot be transported as long distance as the dissolved phase. Please clarify.

Lines 397–402: Concentrations of certain elements may be high in sea ice, but what about their relative mass compared to the surface ocean? In other words, how much mass is incorporated into the sea ice in a budget perspective? The volume of sea ice seems to be much smaller than that of the polar mixed layer.

Lines 592–594: The constants α and a did get mentioned in the reference cited here. However, the authors still need to do a better job explaining these two terms, at least briefly. What do they mean? Just showing their values is not enough.

Lines 603–608: Are there any figures? If using d_{18O} alone generates similar results, what new information do we get by adding an extra Nd tracer in the end-member mixing model?

Lines 780–795: Why are the white arrows just one way? What about sea ice algae falling, dirty sea ice exchange, or ice

melting along the way?

%-----

Editorial remarks (by line #):

Line 1: Is "material" a better word than "matter"? I found the phrase "Siberian matter" a bit strange.

Line 69: Ditto. It's a bit weird to use "matter fluxes" here. Do the authors think other phrases such as "material fluxes" or "elemental fluxes" work better? If so, the authors should consider changing this word/phrase throughout the manuscript.

Line 90: I don't think "the reasons mentioned above" are immediately before this sentence or even in this paragraph. If that's the case, the authors should consider summarizing the reasons briefly and adding them here.

Line 614: I may have missed it, but what is "CO"? Please clarify.

Lines 671–673: The end of the first drift is not labeled by a dashed black circle.

Lines 675–678: What are those dashed arrows? Please clarify.

Reviewer #3

(Remarks to the Author)

This study uses "provenance" tracers (salinity, d18O and eps_Nd) to identify the origin of surface waters and sea ice. They use standard linear deconvolution techniques for surface seawater (<200 m), and also apply the technique to sea ice samples, which is novel. Critically, their sampling campaign (from the MOSAiC project) followed an ice flow through an entire annual cycle. There have been a couple of prior floating platform expeditions to the Arctic, but they came before our capacity to collect and perform isotopic analyses on rare earth and element (REE) samples. Thus, the authors' dataset is truly unique and represents a major leap forward for provenance studies of Arctic waters. Their results use a recently described river tracer, neodymium isotopes, to assess which rivers are contributing to the riverine fraction in their samples. The results indicate an admixture of Lena and Yenisey/Ob waters in the surface ocean and sea ice; suggesting that sea ice incorporates dissolved rare earth constituents in the same abundances that it is in the surface ocean. Further, they show that the balance of Laptev Sea (Lena) vs Kara Sea (Ob/Yenisey) waters shifts in major ways along the route of the platform's drift. And finally, they demonstrate that these shifts are visible in the ice, so that a vertical core through the ice can be viewed as an archive of surface water chemistry. These results have implications for the transport of constituents by sea ice to locations distal from the source.

The submission represents the latest in a series of excellent studies from this group. This latest effort includes a superb dataset and some exciting results. However, we have several serious problems with the presentation that need to be addressed before publication. We recommend accepting the paper, pending major revisions.

General Comments

Early in the text, the authors indicate that prior work mainly views the near-surface waters as traveling in the same direction as, and at about the same speed as, the sea ice. And at several points in the text, the authors indicate that an important part of their contribution is to show that the surface waters travel differentially from the ice. But this has been well known for many years. In fact, later in the text, and in their Figure 2, they indicate that the pre-existing consensus model is that the near-surface waters travel to the right of, and significantly slower than, the sea ice. Thus, the initial characterization of prior work feels to us like a "straw man" argument: setting up an inaccurate version of their colleagues' previous work in order to stand out in comparison.

Previous work has documented an Ekman spiral in Arctic waters, localized areas of intense upwelling between ice floes, eddies of various sizes, geostrophic currents based on horizontal density gradients, topographically trapped boundary currents, and horizontal detrainment from boundary flows, in addition to ice-ocean surface stresses as drivers of near-surface currents. If we take this diversity of dynamics as the actual prior state of knowledge, then the authors' unique contribution is not the fact that surface water and sea-ice motions diverge, but something specific about how they diverge.

Regarding the mechanism of divergence, we had difficulty understanding with some precision what the authors are suggesting. On the one hand, they refer to hydrodynamics on the Siberian shelves as the origin of distinct assemblages of water masses crossed by their floe. On the other hand, they suggest that a variety of short-term, mid-scale phenomena might be responsible, such as inertial motions and eddies. The former hypothesis suggests a relatively stable TPD into which a varying set of "head waters" are inserted. The latter is a different narrative, in which time-varying motions in the pelagic ocean might be responsible for shifting TPD content. Perhaps they envision both: varying sources where the TPD detains from the shelf, followed by insertions and/or detrainments along its route. Whatever the narrative, it needs to be clearer.

We note as well that dissolved tracers have historically been applied to problems that require a long-term integrative metric. This is because they disperse over their residence times, smoothing out short-term, small-scale flows and structures. Current measurements and purposeful tracer injections have been the tools of choice for evaluating dynamics with shorter time

scales. The MOSAIC program included current profilers, and we found it curious that the manuscript made no mention of that data, which should have a lot to say about the divergence of sea ice and water velocities along the track.

A second major outcome of the work is that sea ice transport of marine constituents is an important process. The study identifies that riverine constituents are a component of sea ice; and suggests that melting of the ice transports riverine signals to the central Arctic and beyond. They state that this paradigm is more important than previously thought, but they do not quantify the relative import of this signal (melting sea ice) to the surface ocean for the various constituents. The reader cannot know how important the mechanism is. Could it still be negligible relative to the role of advected waters? Is the bulk of material transported from the ice “factories” of the Siberian shelf to the Greenland and Canadian shelves? Some sort of scale should be set to support the claim of importance. Furthermore, they imply that sea ice as a modality of transport of materials from Siberian shelves has so far been neglected, and do not cite studies that show that sea ice entrains and redistributes materials. We agree that quantification of this process is an ongoing area of study, and this study offers a lens into sea ice formation endmembers that permit a better quantification than previous studies. However, the role of entrainment of geochemical parameters and other material has not previously been neglected. Entrainment of geochemical parameters in proportions present in the surface seawater is consistent with our understanding of sea ice formation (e.g., major geochemical constituents are driven by brine/salinity dynamics). To acknowledge some of the previous work addressing the transport of sea ice incorporated constituents we advise the authors look at: [https://doi.org/10.1016/0048-9697\(95\)04174-Y](https://doi.org/10.1016/0048-9697(95)04174-Y), <https://doi.org/10.1029/2022GB007320>, <https://doi.org/10.1038/srep16179>,

The authors assume that the samples are composed of Yenisey/Ob water, Lena water, sea-ice melt, and Arctic Atlantic water. In the Supplementary Material, they make the case that Pacific inflow is not present in the samples. But what about other freshwater sources, such as Kolyma River water and other Eurasian rivers, groundwater seep, and Mackenzie River water? It seems to us that something needs to be said about why these sources can be ignored.

The only error mentioned in the manuscript is the analytic error on neodymium measurements. However, in our experience, by far the largest source of error in water mass analysis is the scatter in the end member observations. This is a serious gap, and really needs to be addressed.

We have a question about the water mass analysis: To estimate their four unknowns, the authors have four parameters: mass, salinity, oxygen isotopes, and neodymium isotopes. Why not set up a 4-by-4 linear problem, and invert a single matrix for each sample? Rather, they take two steps, first identifying the sea-ice fraction, and then identifying the other three components. What is gained by splitting the process in two?

We are attaching a marked-up pdf file of the manuscript with a number of more specific comments and questions.

Reviewer #4

(Remarks to the Author)

Version 1:

Reviewer comments:

Reviewer #1

(Remarks to the Author)

(Re-)Review of “Dynamic Ice–Ocean Pathways along the Transpolar Drift Amplify the Dispersal of Siberian Matter” by Laukert et al., submitted to Nature Communications

I appreciate the efforts taken by the authors to address the comments of their initial reviewers. I think the ms. has improved significantly from the first version. This new version reads well and is superbly illustrated. I still hold reservations on many aspects of the work, but their arguments are generally all valid. While my original sentiment holds true – that these are interesting data collected from a unique sampling campaign – I still have two reservations. The first is a technical issue: The samples were collected in a lagrangian sense (drifting with the floes), but the data seems to be interpreted in an eulerian sense. I worry about regional interpretations being made from locally collected data in this instance – e.g., how much of their sampling could be simply caused by local ice floe-sea water interaction/evolution versus regional inputs/outputs? My second issue is simply that I was uninspired by their conclusions (a problem I had in the original version). This may not be an issue for publication, as the other reviewers seemed to consider this work important. I defer to the editors and other people more interested in the themes addressed in this work; is this interesting work?

In summary, I think the re-write has helped tremendously, and I could envision this published in Nat. Comms.. Any perceived ambivalence to publication should not be seen as a demerit, rather my own ignorance of the greater science this work addresses.

Reviewer #2

(Remarks to the Author)

The authors have fully addressed my questions and done a good job responding to the points brought up by other reviewers. I am satisfied with the authors' explanations and edits in the revised manuscript. I recommend accepting the manuscript.

Reviewer #3

(Remarks to the Author)

This is a resubmission of a manuscript that I reviewed previously.

All of my comments and criticisms have been fully addressed. In fact, the authors' responses were more detailed and extensive than usual, and so have made interesting reading in themselves.

They have extensively reworked several parts of their argumentation; and they have added significantly to their supplementary material. Changes to the charts and wordings of various sentences and paragraphs have made the manuscript much clearer and easier to read.

I recommend publication of the current manuscript without further revision.

Reviewer #4

(Remarks to the Author)

RESPONSES TO REVIEWER COMMENTS

Line numbers correspond to the finalized, clean version of the document without marked edits.

Reviewer #1:

In this ms., the authors present a suite of novel neodymium and oxygen isotope, rare earth element and salinity data from a unique expedition sampling throughout a winter season in the Arctic ice. Their data are interpreted in terms of input of freshwater from the Siberian rivers to the Transpolar Drift (TPD), an important surface current that crosses the Arctic Ocean, into the highest North Atlantic. Their conclusions are fine, but rather ambiguous – statements that tend to amount to “there is complexity,” but not exactly what this means.

We thank Reviewer 1 for this helpful review and comments and address each of their points below. Following the detailed constructive feedback from Reviewers 3 and 4, we now articulate our conclusions more clearly.

I fear that much of the perceived impact of this paper lies in the unique nature of the sampling campaign, which is an impressive feat. But the data generated and conclusions reached are almost identical to their prior publication in 2021 (Paffrath et al., *Nature Scientific Reports*).

We respectfully disagree with the reviewer on this important point. While the earlier study made notable progress in developing an oxygen and neodymium isotope-based method for quantifying riverine contributions in the Eurasian Basin, our manuscript explores a distinctly different aspect: the redistribution of Siberian matter driven by ocean surface variability and sea ice–ocean interactions. Our research, through the first-ever comparison of sea ice and seawater isotope signatures over a 200-day period, provides critical new insights into the seasonal dynamics of matter transport, a topic that has not been addressed in such detail in previous research, either by Paffrath et al. or by anyone else.

I was hoping that this more recent ms. would have had further, more definitive conclusions on the nature of distribution of riverine sourced tracers, and how they might impact the North Atlantic. Particularly because these data are interesting and I think that they should be published. However, as is, this ms. reads the same as their last (GEOTRACES) work. – perhaps a longer-format journal would be more appropriate for more detailed and deeper discussion of both these data sets?

We appreciate the reviewer’s comments but would like to clarify that our study builds upon, rather than replicates, the work of Paffrath et al. (2021). While their study established a robust framework for tracing river water in the Eurasian Basin using O and Nd isotopes, our research extends this foundation by conducting a quantitative water mass analysis based on these tracers. Specifically, we examine matter dispersal mechanisms through a more comprehensive dataset that encompasses both seasonal water column and sea ice data. This allows us to investigate *key processes* in sea ice *and* the surface ocean. We recognize that the original manuscript may not have sufficiently emphasized these advancements. To address this, we have revised the text to better articulate how our work builds upon and expands the scope of earlier studies (e.g., L94-97).

While we believe that Paffrath et al. have in large parts already addressed the reviewer’s concerns regarding river discharge variability, the influence of Pacific water, and non-conservative processes (points 1-6 raised below), we acknowledge the importance of revisiting these issues in light of our recent observations and studies published after Paffrath et al. Accordingly, we have thoroughly revised and expanded our discussion to explore these issues in greater depth, as outlined in our individual responses below. River discharge variability is now emphasized in the main text, as it plays a crucial role in understanding matter variability. However, since non-conservative processes and Pacific water do not impact our conclusions, we have opted to address them in the supplement. We believe this approach aligns with the journal’s format, as including these topics in the main text would not add significant insights or affect the overall conclusions and their significance.

Regardless of decision, I might offer the authors the following concerns to consider:

1. River water input is highly variable over the year, particularly in the Arctic. The winter has near-negligible discharge, compared to massive spring flows. There is no mention of this in the ms., but I would imagine it is important. Similarly, what is the residence time of river water in the shelf area, or in the TPD?

We recognize that readers unfamiliar with previous studies (e.g., Paffrath et al., 2021, *Sci. Rep.*; Janout et al., 2020, *Frontiers*; Laukert et al., 2017, *EPSL*) may require more background regarding river discharge variability and its impact on the TPD. As we mentioned in the original manuscript, hydrographic variability in the source regions is a key factor contributing to the freshwater distribution encountered along the TPD. To clarify this, we have extended our original discussion of the processes driving this shelf variability by including the influence of highly variable river discharge across seasons, L254-267. We have also refined our water mass analysis by providing an uncertainty estimate for the variability in end-member signatures, L620-623. This estimate, based on a Monte Carlo simulation, now accounts for the slight seasonal differences in tracer signatures, e.g., between winter and summer months for the Lena River (endmember ranges are now provided in Suppl. Table S1).

As for the residence and advection times, Paffrath et al. (2021) compared their TPD data from 2015 with data from earlier years (2013, 2014) from the Laptev Sea (Laukert et al., 2017, *EPSL*) and found that the distribution of tracers in both the TPD and its main source region, the Laptev Sea, aligns well with known residence and advection times of surface waters in this area. For clarity, we now provide relevant residence and advection times in the manuscript, L254-267.

2. The systematics of eNd are oversimplified. This is not an open-ocean setting, and the residence time of Nd is not the same as deep, open ocean (it cannot be). The “quasi-conservative” nature of eNd in this setting cannot be assumed.

We agree that the residence time of Nd in the surface and coastal ocean is shorter than in the deep ocean; however, this difference, which is poorly defined, does not affect our analysis or conclusions. The transport of ice and water along the TPD occurs over relatively short timescales, estimated at less than 10 years (Pfirman et al., 1997, *Sci. Total. Environ.*; Anderson et al., 1994, *JGR Oceans*; Schulz et al., 2024, *Elementa*). Thus, a residence time of hundreds of years, as in the deep open ocean, is not necessary to explain the observed ϵ_{Nd} behavior. This point is now acknowledged and discussed in the main text, L166-173, and Suppl. Text 1.

Furthermore, Laukert et al. (2017, *EPSL*) demonstrate that REE removal, which reduces Nd residence time, is confined to the shelf, a finding supported by our water mass analysis, which shows close alignment between measured and expected [Nd] following removal rates of 70-80% (see our reply to point 4 and new Suppl. Fig. S5). Moreover, invariant vertical REE profiles in the central Arctic Ocean indicate the absence of (reversible) scavenging. This stability has been attributed to suppressed biological productivity and limited particle flux, which restrict REE cycling (Paffrath et al., 2021, *JGR Oceans*) and associated non-conservative changes in dissolved ϵ_{Nd} (see our reply to point 6). This is now also mentioned in Suppl. Text 1.

3. I would not be shocked to see 75% Pacific water at these depths. Saying it is “improbable” seems suspect. Pacific water penetrates very far into the Arctic, particularly in the winter when river flows are very low. Again, see point 1.

We regret our initial wording was strong and appreciate the opportunity to clarify. However, to the best of our knowledge, no study convincingly demonstrates significant penetration of Pacific water into the deep Eurasian Basin. The reviewer's assertion is thus not supported by literature. The perception of extensive Pacific water distribution in the Eurasian Arctic Ocean may arise from older studies based on nutrient relationships, which have since been shown to be unreliable for tracing Pacific water (see summary in Suppl. Text 2.4). In fact, recent studies based on standard hydrographic data demonstrated that only minor fractions of Pacific water (<30%, Lin et al., 2021, *GRL*) are present in the form of summer Pacific water in the uppermost layers (<100 m) of the Canada Basin and that this water is separated from the Eurasian Basin by a front along the northern side of the Mendeleev and Alpha ridges (Planat et al., in review at *JGR Oceans*). Even if we assume that some Pacific water escapes this circulation regime and enters the Eurasian Basin, it would undergo significant dilution during transport, likely reducing fractions to well below 30% in the upper 100 m. However, our data suggest that even this scenario is unlikely for our sites (see response to point 5). We have now expanded our rationale for excluding Pacific water from our water mass analysis in the supplement (Suppl. Text 2.4), incorporating the above arguments and references.

4. I do not trust any of the [Nd] arguments. There is nothing conservative about [Nd] to suggest it can be used in these ways – particularly when the “mixing” plot already assumes a 75% loss in the measured riverine end member (i.e., most of the Nd is lost in the estuary, but how consistently?). That they found a linear mixing curve is interesting, to be sure (dependent on some assumptions). But it makes little sense with respect to the isotopes; how can the concentration continue to increase with apparent equal sensitivity/change in the isotopic fingerprint? The downstream isotopes should not change as readily, unless there is an effective scavenging, which calls into question the [Nd] patterns.

The reviewer raises concerns regarding an apparent inconsistency between the strong correlation observed between [Nd] and river water fraction ($R^2 = 0.9$) and the simultaneous change in ϵ_{Nd} in our data (Figs 4a and b). As argued before, the ϵ_{Nd} shift can be explained by gradual admixture of river water to AAW, initially from the Yenisei/Ob rivers and subsequently from the Lena river. The persistence of increasing [Nd] despite the changing ϵ_{Nd} can be attributed to the similar [Nd] concentrations but differing ϵ_{Nd} of the two riverine endmembers (Suppl. Table S1), combined with differences in their densities (and thus river water fractions) due to their respective trajectories across the continental shelves. We have incorporated this information to the main text, L176-184 and L216-221. Paffrath et al. (2021, *Sci. Rep.*) provide compelling evidence for this interpretation, demonstrating, through the consideration of advection times, near-identical tracer properties between the source regions of the waters and the corresponding TPD river water domains, confirming that a change in river water origin drives the ϵ_{Nd} shift.

Furthermore, we do not include [Nd] in our water mass analysis, which allows us to compare the calculated [Nd] values resulting from the conservative mixing of the selected endmembers with the measured [Nd] values. These data closely align (new Suppl. Fig. S5), supporting a mixing relationship. We have detailed this comparison now in Suppl. Text 1 and referenced it in the ‘Methods’ section. The Nd removal process necessary for this mixing has been well characterized in the Laptev Sea (Laukert et al., 2017, *EPSL*), where removal rates exceed 70% for surface waters exported from the Laptev Sea ($S > 25$). We now acknowledge the limited variability in estuarine Nd removal (70-80%) and have incorporated an uncertainty estimate for this process in our water mass analysis through a Monte Carlo simulation, which is mentioned in the main text (L186-188), the ‘Methods’ section (L618-620) and Suppl. texts 1 and 2.2.

The scavenging process proposed by the reviewer appears less plausible than our interpretation for explaining the observed patterns due to several key inconsistencies. First, it does not account for the systematic shift to less radiogenic ϵ_{Nd} values at $f_{RIV} > 10\%$ while maintaining a *consistent and robust* [Nd]- f_{RIV} relationship. Second, it fails to address the origin of the more radiogenic signatures observed at intermediate f_{RIV} reaching -9, especially given the dominance of AAW in this region, which has ϵ_{Nd} values no more positive than -10. Lastly, under the scavenging scenario, a shift to less radiogenic values at $f_{RIV} > 10\%$ would correspond with *decreasing* [Nd] ($AW < 16$ pmol/kg), rather than the observed steady increase in concentrations.

5. I want to see the Pacific treated quantitatively in their ms., and shown to be invalid, in figures and calculations. They did this in their 2021 publication, but did not here. In neither publication is a $1/[Nd]$ vs. ϵ_{Nd} plot shown, which should, if conservative, show clear mixing patterns.

Paffrath et al. *did* include a $1/[Nd]$ vs. ϵ_{Nd} plot, albeit in their supplement to illustrate mixing patterns (their Fig. S1). However, we opted not to include this type of plot in the present study because the observed mixing patterns remain ambiguous. The nearly identical ϵ_{Nd} signatures of Pacific water and Yenisei/Ob water render these patterns inconclusive (as discussed by Paffrath et al.). We also recognize the importance of quantitative analysis in assessing Pacific water influence. However, the 4-component mixing calculations presented by Paffrath et al. are inherently sensitive to small variations in the $[Nd]$ of riverine endmembers. Our sensitivity tests confirm that even minor changes in riverine Nd concentrations significantly impact the calculated fractions, primarily through the Nd concentration balance. This limitation, which Paffrath et al. also acknowledged (Supplementary Text: “*Limitations of the water mass inventory based on the Nd-method*”), makes 4-component assessments with Pacific water as one of the endmembers unreliable.

We have therefore adopted a different approach, now refined further. We first incorporated recent findings from other studies on the spatial Pacific water distribution in the Arctic Ocean (see response to point 3). We then replaced the Yenisei/Ob component in our analysis with Pacific water to determine the volume required to explain the more radiogenic ϵ_{Nd} signatures observed in the surface to ~100 m depth. This analysis indicated that an unrealistically large volume of Pacific water—contradicting recent studies—would be required to account for the observed signal, thereby supporting its exclusion from our study site. Additionally, even if Pacific water were hypothetically present below 60 m depth (a highly unlikely scenario based on current data), it would not alter our conclusions about sea ice–ocean interactions and surface ϵ_{Nd} variability, given that the surface ϵ_{Nd} signal would still predominantly reflect Yenisei/Ob contributions. We added this expanded discussion, along with supporting evidence against the presence of Pacific water, to a new section in the supplement (Suppl. Text 2.4).

6. The boundary exchange problem for ϵ_{Nd} must be addressed. Boundary exchange has been shown to not impact $[Nd]$ from original observations. The authors dismiss this citing lack of evidence in overlying water, but this is weak, and was indeed weak in the original boundary exchange publications.

Thank you for this suggestion. We acknowledge the need for a more robust discussion on this topic. In our initial submission, we had briefly addressed the influence of seawater-particle interactions on ϵ_{Nd} in a paragraph within Supplementary Text 1, mainly referring to earlier studies. However, we have now expanded this discussion into a dedicated section (new Suppl. Text 1). In this new section, we present several arguments to illustrate why boundary exchange or REE release from particles is unlikely to occur in our study region. These arguments include: the absence of observed processes to date (supported by five independent studies: Laukert et al., 2017, *GCA*; Laukert et al., 2017, *EPSL*; Laukert et al., 2019, *Chem. Geol.*; Paffrath et al., 2021, *Sci. Rep.*; Paffrath et al., 2021, *JGR Oceans*), limited biological activity due to perennial ice cover, minimal riverine particle flux, and high ratio of organic to inorganic nanoparticles and colloids in Siberian rivers. Only one sample—which we have excluded from our source apportionment analysis—could potentially be influenced by such processes. We have also revised the introduction and results sections to acknowledge potential non-conservative influence, see L86-88 and L169-171.

7. Brine spillage – I appreciate the difficulties of sample collection, and this should not be held against the authors, but should be treated as honestly as possible. The fact that brine has far greater $[Nd]$ concentrations than non-brine water makes a linear mixing plot even more confusing. This should be treated with greater discussion.

We appreciate the reviewer’s concern, but we are unsure how brine *spillage* specifically during ice core retrieval would affect the $[Nd]$ - f_{RIV} mixing plot. If the reviewer’s comment refers instead to brine *rejection*—that is, the influence of sea ice formation on water column $[Nd]$ —we acknowledge that this was not fully addressed in our initial water mass analysis. We have now updated the analysis to account for sea ice processes, see L599-608. This adjustment slightly changed fractions but did not alter our conclusions. Please also see our extended response to Reviewer 2 for additional context.

8. L68-69 needs references? This seems like a bold statement, given Ekman transport is one of the original “oceanographic” theories, based on years of observation and is a cornerstone of physical oceanography. (The story of Nansen is the oceanography equivalent to Newton’s apple...). Throughout the ms., bold statements confronting established physical oceanographic principles should be considered deeper.

Our intention was not to question basic principles of physical oceanography. Instead, we focused on assessing the combined impact of Ekman transport and other processes on the dispersion of Siberian matter, which is a more nuanced issue. We have now revised the manuscript following the detailed constructive feedback from Reviewers 3 and 4 on this topic.

In summary, I feel that there are a few critical points that should be addressed regardless of where this ms. is published. I do hope these data are published, and it would be great if published with Nature Communications - but I think this would take a significant re-write.

We appreciate the reviewer’s support for publishing our data and the encouragement for eventual publication in *Nature Communications*. In response to this and the other reviews, we have made substantial revisions to the manuscript.

Reviewer #2:

Laukert et al. present a well-written manuscript presenting geochemical provenance tracer data collected from the MOSAiC campaign in the Eurasian Arctic Ocean. This exciting dataset is potentially significant because it fills the knowledge gap of sea ice and ocean surface pathways in the polar winter. The manuscript is clear, thorough, and nearly free of typos. The authors have done a good job combining multiple chemical tracers and disentangling complex endmembers. However, I have some substantive comments, as listed below. One of my main concerns is that the authors may have mistakenly omitted the sea ice melt fraction in their mass balance Equation 3, which could generate quite different results from the current version, at least during part of the cruise. Overall, I recommend publication with major revisions.

We would like to thank the reviewer for this positive assessment and the constructive review. The main issue raised by the reviewer regarding the sea ice fraction, as well as all other issues, have been thoroughly addressed as described below.

Major interpretation points

This reviewer has some major questions about Equations 3 to 5. First, the authors should add f_{ice} in Equation 3 if I correctly understand Rosen et al. (2015)'s Equation 3. Without f_{ice} , the sum of f_{AW} , f_{Lena} , and f_{YenOb} wouldn't be 1. Secondly, the authors correct for salt fluxes related to sea ice formation and melting for salinity in Equation 4, but no such correction exists for Equation 5. The authors could argue that neodymium isotopes remain unfractionated during sea ice processes, but sea ice processes do contribute to Nd concentrations, as seen in Figure 5d. This would change the denominator of Equation 5. Thirdly, if my previous two points are correct, I am curious how these fractions would change after taking into account the fraction of sea ice melt in Equations 3 and 5. I don't think the general story would change, but there could be some changes to some interpretations, at least during part of the cruise.

We appreciate the reviewer's thorough analysis of our water mass calculations. Initially, we chose to exclude the sea ice fraction (f_{SIM}) from our analysis to focus solely on water mass origin, without considering sea-ice influence. However, we agree with the reviewer that including f_{SIM} , as well as correcting for [Nd] in relation to sea ice processes, is indeed relevant and valuable. Consequently, we have revised our water mass analysis to include the sea ice fraction and $\delta^{18}O$ balance in the matrix inversion, thereby also addressing suggestions from Reviewers 3 and 4. To account for uncertainties in endmember values, the matrix inversion was performed using a Monte Carlo simulation with 100,000 iterations. See 'Methods', L599-632.

Importantly, as expected by the Reviewer, the overall fractions and patterns did not change significantly from our original results. This is largely because the key impact of sea ice—its effect on salinity—was already accounted for in our initial calculations. Furthermore, the influence of the [Nd] correction on the fractions appears to be limited, presumably because of the very high endmember [Nd] of the rivers. However, the river fraction has increased slightly due to refinements in endmember values and updates to the $\delta^{18}O$ dataset, which is now complete and has been quality-checked. The calculated f_{SIM} values align with previous studies, indicating significant brine addition to the upper 100 m and small amounts of melt beneath 100 m. We have added this information to the revised manuscript, L154-163. Please note that a detailed discussion of the f_{SIM} distribution is outside the scope of this study, though we have plans to address this topic in a follow-up separate publication.

Interpretation points, by line

Lines 31–32: The decoupling between sea ice and ocean surface transport doesn't seem to only occur in the winter. This part of the introduction has been completely revised and no longer contains this statement.

Lines 50–51: The Fram Strait is not the only outflow of the TPD. What about the outflow into the Canadian Arctic Archipelago (CAA)? For example, the authors can check out their own schematic Figure S3 or Carmack et al. (2016) (10.1002/2015JG003140).

We agree and included the CAA as a potential exit region of TPD waters and sea ice, L50-52. We also added the reference provided by the reviewer to the manuscript, making it the first reference in the main text.

Lines 206–208: The authors should consider plotting $d_{18}O$ and Nd section plots in the supplemental. It's not immediately obvious if the Pacific water has some influence on the subsurface water mass, especially for those close to the Makarov Basin, which can be identified by Paffrath et al. (2021). Additionally, how would the Greenland glacier runoff influence ($\epsilon_{Nd} < -17$; Figure S3) the end-member mixing model close to the Fram Strait?

We have added these plots to the supplement (Suppl. Fig. S1), but unfortunately, they do not provide new insights into the influence of Pacific water, as the underlying issue with tracer sensitivity to Pacific water remains unchanged (see Paffrath et al., 2021, *Sci. Rep.*, and our reply to point 5 of Reviewer 1). However, we note that our data do not exhibit the deviations in the [Nd]- f_{RIV} relationship observed by Paffrath et al. in the Makarov Basin. This is shown in our new Fig. 4 (with recalculated f_{RIV} values for the Paffrath dataset) and discussed in the main text (L176–180). These deviations may reflect Pacific water input, but this is irrelevant to our study. We added this point to a new, extended discussion on Pacific water influence in the supplement (new Suppl. Text 2.4), which also includes new references, as detailed in our response to point 3 of Reviewer 1.

Indeed, we observed some influence of Greenland meltwater in one sample from Fram Strait. This sample was excluded from our water mass analysis, as detailed in the *original* manuscript (L720-L723) and now mentioned in L173-176. We have included a deeper discussion about the potential contribution of this endmember in the supplement (Suppl. Text 2.2).

Lines 335–337: It's hard to reach this conclusion with such a limited number of Nd or REE measurements as compared to salinity and $\delta^{18}\text{O}$. I am also not sure why the authors have different conclusions between the pattern of ϵ_{Nd} (e.g., Lines 292–295) and Nd or HREE/LREE. All of those parameters only have 5 points in Figure 5. Are there any statistical reasons? We acknowledge that five data points are generally not considered statistically significant. The low number of samples results from the more challenging sampling and processing requirements for large-volume ϵ_{Nd} samples, as well as the limited sampling capacities during MOSAiC. Nevertheless, the observed changes in the surface water and their relationship with sea ice data are *clear* and *distinguishable* from measurement uncertainties. Additionally, these trends align with existing knowledge about REE systematics in sea ice (Laukert et al., 2022, *GRL*), supporting the validity of our interpretation.

Despite having the same quantity of seawater data, our conclusions regarding the comparability of ice and water differ due to the lower resolution of ϵ_{Nd} measurements in sea ice (20 cm intervals) compared to Nd, HREE/LREE, salinity, and $\delta^{18}\text{O}$ (10 cm intervals). We want to emphasize that the *original* text (L292–295) discusses the impact of an adjustment of ϵ_{Nd} due to permeability effects, which we conclude would be less evident than in salinity/ $\delta^{18}\text{O}$ due to the 20 cm resolution. We have now refined this part of the manuscript (L324-331) and hope that this adequately addresses the reviewer's comment.

Line 344: Why does this have implications for particles? All parameters discussed in the manuscript are dissolved. Particles tend to sink, so they cannot be transported as long distance as the dissolved phase. Please clarify.

Thank you for bringing attention to this important point and the opportunity to clarify. Broadly speaking, the transition from dissolved to particulate phases is gradual rather than discrete; the spectrum spans from ionic solutions to nanoparticulates, ligand-associated compounds, colloidal matter, and small particles. Many of these size classes, including smaller particles, are neutrally buoyant and can therefore be transported over considerable distances as suspended particulate matter (SPM). For example, natural SPM linked to river input has shown to travel hundreds of kilometers away from the river mouth (e.g., Wegner et al., 2013, *BG*; Yu et al., 2023, *Mar. Environ. Res.*; Brown et al., 2014, *JGR Oceans*). This transport can be amplified similarly to the dissolved load when sea ice is involved through multiple freezing/thawing cycles. In addition, some particles always stay at the ocean surface (so called floaters, e.g., certain plastic particles). Therefore, our assessment of potential transport routes considers the longest possible distances that both dissolved substances and particles can cover. We have now clarified this in the manuscript, albeit in a very condensed form due to the length restriction, L407-410.

Lines 397–402: Concentrations of certain elements may be high in sea ice, but what about their relative mass compared to the surface ocean? In other words, how much mass is incorporated into the sea ice in a budget perspective? The volume of sea ice seems to be much smaller than that of the polar mixed layer.

We appreciate the reviewer's point regarding the relative mass of elements incorporated into sea ice compared to the surface ocean. Indeed, our objective was not to suggest that this process significantly influences the overall Nd or REE mass balance in the Arctic. Instead, our study aims to demonstrate that Nd and its isotopes can serve as valuable tools for examining redistribution processes and identifying *potential* transport pathways. While the impact of sea ice melt on REE levels in the mixed layer is likely minor, it is important to note that this primarily applies to elements present naturally in *trace amounts*.

Apart from that, the redistribution behavior varies significantly among substances due to their distinct physical and chemical properties, which is now acknowledged early on (L60–63). For example, substances such as floaters often remain confined to the surface rather than mixing throughout the mixed layer, suggesting they may be fully redistributed by sea ice. Additionally, variability in source inputs, particularly for anthropogenic pollutants, complicates precise flux estimations, see e.g. microplastics, which exhibit extremely high source variability (e.g., Peeken et al., 2018, *Nat. Commun.*). These complexities underscore the challenges of deriving accurate budgets based on observed distributions, such as those of REEs. Therefore, we rather emphasize the importance of the underlying transport mechanisms, which we now clearly articulated, L410–415.

Lines 592–594: The constants α and β did get mentioned in the reference cited here. However, the authors still need to do a better job explaining these two terms, at least briefly. What do they mean? Just showing their values is not enough. This passage has been revised and for the calculation of S_0 we now refer entirely to Rosen et al. 2015, L676-678.

Lines 603–608: Are there any figures? If using $\delta^{18}\text{O}$ alone generates similar results, what new information do we get by adding an extra Nd tracer in the end-member mixing model?

Thank you for pointing this out; we agree that clarification was needed. Incorporating ϵ_{Nd} allows us to differentiate between Lena and Yenisei/Ob waters, which is not possible with $\delta^{18}\text{O}$ alone. While $\delta^{18}\text{O}$ enables calculation of the combined river contributions (f_{RIV}), allowing us to examine these contributions at a higher spatial resolution (Fig. 3a), ϵ_{Nd} provides a coarser but more distinct separation (Fig. 3c-e). This complementary approach enhances our understanding of riverine inputs and their distribution. We have revised the main text (L184-186) and the methods section (L633-639) to clarify this distinction.

Lines 780–795: Why are the white arrows just one way? What about sea ice algae falling, dirty sea ice exchange, or ice melting along the way?

We wanted to emphasize the *dominant* redistribution mechanisms for river-borne substances, which we now clarified in the figure caption, L839-840.

Reviewer #3:

This study uses “provenance” tracers (salinity, d18O and eps_Nd) to identify the origin of surface waters and sea ice. They use standard linear deconvolution techniques for surface seawater (<200 m), and also apply the technique to sea ice samples, which is novel. Critically, their sampling campaign (from the MOSAiC project) followed an ice flow through an entire annual cycle. There have been a couple of prior floating platform expeditions to the Arctic, but they came before our capacity to collect and perform isotopic analyses on rare earth and element (REE) samples. Thus, the authors’ dataset is truly unique and represents a major leap forward for provenance studies of Arctic waters. Their results use a recently described river tracer, neodymium isotopes, to assess which rivers are contributing to the riverine fraction in their samples. The results indicate an admixture of Lena and Yenisey/Ob waters in the surface ocean and sea ice; suggesting that sea ice incorporates dissolved rare earth constituents in the same abundances that it is in the surface ocean. Further, they show that the balance of Laptev Sea (Lena) vs Kara Sea (Ob/Yenisey) waters shifts in major ways along the route of the platform’s drift. And finally, they demonstrate that these shifts are visible in the ice, so that a vertical core through the ice can be viewed as an archive of surface water chemistry. These results have implications for the transport of constituents by sea ice to locations distal from the source.

The submission represents the latest in a series of excellent studies from this group. This latest effort includes a superb dataset and some exciting results. However, we have several serious problems with the presentation that need to be addressed before publication. We recommend accepting the paper, pending major revisions.

We would like to express our sincere gratitude to Reviewers 3 and 4 for their thorough and insightful reviews, as well as for their positive remarks on our study—noting it as a major advancement in provenance studies of Arctic waters! We deeply appreciate the time and effort invested in providing such detailed, constructive feedback. Reviews with this level of engagement are invaluable, and we are grateful for the opportunity to enhance our work accordingly.

As outlined below, we have carefully addressed all points raised and made substantial improvements to the manuscript. While most feedback was clear and actionable, a few comments appeared to arise from minor misunderstandings, which we have clarified to better communicate our findings.

General Comments

Early in the text, the authors indicate that prior work mainly views the near-surface waters as traveling in the same direction as, and at about the same speed as, the sea ice. And at several points in the text, the authors indicate that an important part of their contribution is to show that the surface waters travel differentially from the ice. But this has been well known for many years. In fact, later in the text, and in their Figure 2, they indicate that the pre-existing consensus model is that the near-surface waters travel to the right of, and significantly slower than, the sea ice. Thus, the initial characterization of prior work feels to us like a “straw man” argument: setting up an inaccurate version of their colleagues’ previous work in order to stand out in comparison.

Previous work has documented an Ekman spiral in Arctic waters, localized areas of intense upwelling between ice floes, eddies of various sizes, geostrophic currents based on horizontal density gradients, topographically trapped boundary currents, and horizontal detrainment from boundary flows, in addition to ice-ocean surface stresses as drivers of near-surface currents. If we take this diversity of dynamics as the actual prior state of knowledge, then the authors’ unique contribution is not the fact that surface water and sea-ice motions diverge, but something specific about how they diverge.

Regarding the mechanism of divergence, we had difficulty understanding with some precision what the authors are suggesting. On the one hand, they refer to hydrodynamics on the Siberian shelves as the origin of distinct assemblages of water masses crossed by their floe. On the other hand, they suggest that a variety of short-term, mid-scale phenomena might be responsible, such as inertial motions and eddies. The former hypothesis suggests a relatively stable TPD into which a varying set of “head waters” are inserted. The latter is a different narrative, in which time-varying motions in the pelagic ocean might be responsible for shifting TPD content. Perhaps they envision both: varying sources where the TPD detrains from the shelf, followed by insertions and/or detrainments along its route. Whatever the narrative, it needs to be clearer.

We appreciate the reviewers’ insights and acknowledge that our initial discussion of previous work may have unintentionally appeared as a “straw man” argument. It was not our intent to misrepresent earlier studies, and we regret that our focus on certain analytical and technical aspects led to an incomplete presentation of our key findings.

We now emphasize prominently that the existing literature already documents a variety of dynamic processes influencing ice and ocean transport, L63-67. As the reviewers correctly pointed out, the novelty of our study lies not merely in identifying the divergence between sea ice and surface water motions, but in detailing *how* this divergence occurs and, specifically, *how it affects Siberian matter dispersal over long distances*—an aspect that has not yet been explored in such detail. We propose that shelf hydrodynamics drive the observed TPD variability, while the influence of short-term, intermediate-scale phenomena is limited, L234-288 and L388-393.

Although this point is more conceptual in nature, we believe the substantially revised manuscript now conveys this narrative more effectively across all sections, which should become evident upon reading.

We note as well that dissolved tracers have historically been applied to problems that require a long-term integrative metric. This is because they disperse over their residence times, smoothing out short-term, small-scale flows and structures. Current measurements and purposeful tracer injections have been the tools of choice for evaluating dynamics with shorter time scales. The MOSAIC program included current profilers, and we found it curious that the manuscript made no mention of that data, which should have a lot to say about the divergence of sea ice and water velocities along the track.

Thank you for bringing up this important distinction between the use of tracers and other tools for studying circulation changes. We have now clarified the role of tracers in resolving long-range transport in the introduction (L81–83). In our initial submission, we referred to sea ice and surface current velocities, as well as their predominant directions, in the ‘*Context of the MOSAIC Campaign*’ section. However, we chose not to explicitly mention the current profiler data from which these velocities were derived. We have since refined this passage to state: “*Ice drift velocities frequently exceeded surface current velocities, with the predominant ice drift direction often diverging from the underlying water movement*” (L120–121). While the divergence captured in these data is valuable, our primary focus is on understanding how this divergence impacts *large-scale matter dispersal*, where tracers provide more suitable insights than current measurements, as also noted by the reviewers. For a more detailed analysis of the variations in ice drift, surface current velocities, and their predominant directions, we now refer readers to Rabe et al. (2024, *Elementa*) and Schulz et al. (2024, *Elementa*) (e.g., Figs. 15a,d and 16a,b in Rabe et al., 2024; Fig. 5a in Schulz et al., 2024).

A second major outcome of the work is that sea ice transport of marine constituents is an important process. The study identifies that riverine constituents are a component of sea ice; and suggests that melting of the ice transports riverine signals to the central Arctic and beyond. They state that this paradigm is more important than previously thought, but they do not quantify the relative import of this signal (melting sea ice) to the surface ocean for the various constituents. The reader cannot know how important the mechanism is. Could it still be negligible relative to the role of advected waters? Is the bulk of material transported from the ice “factories” of the Siberian shelf to the Greenland and Canadian shelves? Some sort of scale should be set to support the claim of importance.

Thank you for raising this important point. As we noted in our response to Reviewer 2, our focus is not on asserting that sea ice redistribution significantly impacts the overall Nd/REE mass balance in the Arctic. Rather, our goal is to illustrate how Nd isotopes can serve as a useful tracer for studying redistribution processes and identifying potential transport pathways, rather than for providing precise budgets of each substance. Contaminants such as microplastics show highly variable concentrations, which complicates reliable predictions regarding their redistribution. Even if quantitative budgets for certain substances could be determined, they would still fall short of capturing the shifts in transport dynamics that might arise from changing source inputs. The significance of this mechanism also varies based on the unique physical and chemical properties of the transported substances. To address this point, we made it clearer that our aim is not to provide exact quantitative budgets—which would be challenging and potentially impractical—but to shed light on the redistribution processes and their significance within the broader context. We have clarified and expanded on this point in the revised manuscript (L401–415).

Furthermore, they imply that sea ice as a modality of transport of materials from Siberian shelves has so far been neglected, and do not cite studies that show that sea ice entrains and redistributes materials. We agree that quantification of this process is an ongoing area of study, and this study offers a lens into sea ice formation endmembers that permit a better quantification than previous studies. However, the role of entrainment of geochemical parameters and other material has not previously been neglected. Entrainment of geochemical parameters in proportions present in the surface seawater is consistent with our understanding of sea ice formation (e.g., major geochemical constituents are driven by brine/salinity dynamics). To acknowledge some of the previous work addressing the transport of sea ice incorporated constituents we advise the authors look at: [https://doi.org/10.1016/0048-9697\(95\)04174-Y](https://doi.org/10.1016/0048-9697(95)04174-Y), <https://doi.org/10.1029/2022GB007320>, <https://doi.org/10.1038/srep16179>.

We appreciate the reviewer’s suggestion and fully acknowledge the well-established role of sea ice in transporting constituents from the water column. This is now reflected in the introduction, which early on emphasizes sea ice as a transport mechanism and incorporates one of these references along with similar ones (L52–54). However, the point we wanted to make was that most prior studies have primarily focused on the transport of matter from coastal or shelf regions into the deep Arctic Ocean or the North Atlantic. This has contributed to the prevailing view of the TPD as a largely direct transport pathway from the Siberian shelves to Fram Strait, with minimal redistribution or interaction *en route* (e.g., Krumpen et al., 2019, *Sci. Rep.*). Our study seeks to extend this understanding by offering new insights into the *provenance and processes* affecting sea ice-borne matter, specifically highlighting how ice-ocean exchanges and surface layer variability facilitate further redistribution *along* the TPD pathway. We have now clarified this distinction in the revised ‘*Implications*’ section of the manuscript, L410–433. We believe this redistribution mechanism will likely intensify as the Arctic transitions toward a seasonally ice-free state, L438–440.

The authors assume that the samples are composed of Yenisey/Ob water, Lena water, sea-ice melt, and Arctic Atlantic water. In the Supplementary Material, they make the case that Pacific inflow is not present in the samples. But what about other freshwater sources, such as Kolyma River water and other Eurasian rivers, groundwater seep, and Mackenzie River water? It seems to us that something needs to be said about why these sources can be ignored.

We appreciate the opportunity to clarify this point further. Including additional freshwater sources would likely underscore our findings on the dynamic nature of redistribution. While contributions from smaller *Siberian* rivers cannot be quantified in our analysis, we exclude significant input from *Canadian* rivers, including the Mackenzie River, as their discharge is primarily directed into the Canadian Arctic Archipelago. To address these considerations, we have added a new section to the endmember discussion in Suppl. Text 2.2.

The only error mentioned in the manuscript is the analytic error on neodymium measurements. However, in our experience, by far the largest source of error in water mass analysis is the scatter in the end member observations. This is a serious gap, and really needs to be addressed.

We appreciate this valuable observation and agree with the importance of addressing endmember variability. To address this issue, we have conducted a Monte Carlo simulation with 100,000 iterations to thoroughly account for uncertainties in endmember values (see 'Methods' section). Although these uncertainties are notable (e.g., see Fig. 6), they do not significantly affect the overall interpretations. We have included the ranges of endmember values used in the simulation in the Supplement (Suppl. Table S1).

We have a question about the water mass analysis: To estimate their four unknowns, the authors have four parameters: mass, salinity, oxygen isotopes, and neodymium isotopes. Why not set up a 4-by-4 linear problem, and invert a single matrix for each sample? Rather, they take two steps, first identifying the sea-ice fraction, and then identifying the other three components.

What is gained by splitting the process in two?

We appreciate this insightful question. As noted in our response to Reviewer 2, our initial approach of separating the sea-ice fraction from the other components allowed us to address the unique influence of water mass *origin* independently. However, we recognized the value in directly integrating the sea-ice fraction (f_{SIM}) within the broader water mass analysis, consistent with prior studies. This updated approach incorporates both the $\delta^{18}O$ mass balance and a correction for [Nd] redistribution originating from sea ice, as emphasized by Reviewer 2.

Following the reviewers' suggestion, we revised our methodology to use a single matrix inversion approach for the four parameters. This adjustment, along with a Monte Carlo simulation to account for endmember uncertainties, is detailed in the methods section (L599-632). Despite these refinements, the primary water mass fractions and patterns remained similar to our original results, with only minor adjustments due to refined endmember values and updates to the $\delta^{18}O$ dataset, now including the additional quality-checked *Leg 5* data (September 2020). The river water fraction, now calculated using our updated analysis rather than previously published fractions from Paffrath et al. (2021), showed a small change (Fig. 4).

Remarks copied from the marked-up PDF (by line #):

L54-56: This may overstate the case.

We have revised this passage to present a more moderate tone, now substantiated with more appropriate references. L54-57.

L61: "incorporates" rather than absorbs?

No longer relevant here but changed everywhere else as suggested.

L72-74: Is this true? May be for some ... but I'd be surprised to find that it's a generally true statement.

We have revised this paragraph and statement following the major comment. L72-76.

L104: This is not intuitive and might require more explanation.

We rephrased this sentence, see L100-102.

L118: This is the first time I've seen this usage--describing new growth on an old floe as FYI. I like it; just saying: has it become common? Or is this a novel usage?

We believe this term was introduced by Angelopoulos et al. (2022, *Front. Earth Sci.*).

L135: citation needed for depth

No longer relevant as text was changed.

L136-137: To what depth are we talking about here?

We meant the *partial* admixture of AAW to all upper layers. While no longer relevant here due to changes in the text, this is clarified elsewhere, see L152-154.

L139: Was there a period when there was no "relatively fresh lens"? Maybe a sentence defining what this means in this context. Also, is this relative to a history of surface water data from the region? Or relative to neighboring sites?

We have adjusted the wording and now refer to low-salinity water, see L133-134.

L150-151: By 'align', do you simply mean 'in the same direction'? I ask because, given what we all learned in school about the Ekman spiral ... the surprising thing is when they do align. Maybe this is all explained in detail below.

We agree and adjusted the sentence accordingly. L120-121.

L156-162: The authors seem to have written the manuscript as though it were clear that the river water on the Eurasian side of the Arctic basin, including the area between the North Pole and the western Fram Strait gets no, or only negligible, contributions from the Canadian side of the basin. In the supplementary text, they make a convincing case that there is little Pacific water in their samples. But I'm not self-evident to me that absence of a Pacific layer means that the negative d_{18O} signal is entirely from Siberian runoff (and not from the Mackenzie, from Pacific inflows modified on the Chukchi shelf, or snow melt). Mightn't the surface circulation drag Mackenzie water into the TPD?

We agree that more clarity was needed here. As mentioned in our reply to the corresponding major comment, Suppl. Text 2.2 now discusses the potential contributions of Canadian rivers and smaller Siberian rivers. There, we argue based on literature that even major Canadian rivers like the Mackenzie River do not significantly contribute to the TPD.

In terms of modified Pacific water, in Suppl. Text 2.4, we tested for the presence of Pacific water by using the endmember ϵ_{Nd} signature of the Pacific water emerging from the Chukchi Shelf (Suppl. Table S1), and thus account for this modified version. The strong correlation between f_{RIV} and dissolved [Nd], alongside the observed patterns in $f_{RIV}-\epsilon_{Nd}$ space, further rules out a significant contribution from snowmelt (precipitation). Snowmelt would introduce greater variability in these relationships, as [Nd] and ϵ_{Nd} are more heterogeneous in snow and differ from river water signatures. It would also lead to other REE patterns, as shown by the snow pattern in our study (Suppl. Fig. S2) and in Laukert et al., 2022 (GRL).

L177: All of them? Isn't there a (varying degree of) particle scavenging, some mineralogic and some biogenic? Are you saying that Nd and other REEs are conservative away from the shelf seas? Need some clarity here; the statement as it is too broad.

We refined this passage in response to the comments of Reviewer 1, see L166-176. As a result, we have also included a dedicated section in the supplement (Suppl. Text 1) discussing the influence of non-conservative processes on dissolved [Nd] and ϵ_{Nd} . In brief, we demonstrate that only estuarine REE removal affects concentrations beyond conservative water mass advection and mixing. However, this can be corrected for in the water mass analysis by adjusting the river endmember [Nd].

L183-184: I had to read this a few times and I'm still not sure I understand the meaning. When you say continental input, what is the boundary of that (does continental slope sediments count? or just terrigenous runoff?). Is there a known residence time of Nd and the HREE/LREE components relative to a specific process that sets a timeline or age limit on these observations? HREE/LREE ratio is pretty niche and although the reader may be directed to the supplementary material, a sentence introducing the parameter might help the reader digest this information.

We understand that this description may be confusing for readers not familiar with REE systematics in the oceans. To improve clarity, we have expanded this section to include more detailed information on the processes involved and where they occur. The comparison with AAW makes clear that the river water carries a younger "continental signal" relative to ambient ocean water. L228-233.

L184: So: not entirely conservative?

Correct, they are not entirely conservative. Non-conservative influence is now mentioned in the main text (L86-88 and L168-171) and addressed in Suppl. Text 1.

L186-187: Are the signals of the Lena and Yenisey/Ob indicative of rivers on east and west of some location (e.g., Severnaya Zemlya) or are there other rivers (minor and major) from the Siberian sector that could contribute to this?

The endmember values of these rivers represent pure river water signals, which remain unchanged in the ocean beyond conservative dilution through water mass mixing (i.e. mixing with AAW). Therefore, they are unsuitable for distinguishing between water advected east or west of specific locations, such as Severnaya Zemlya. However, we have considered the potential contributions from other Siberian rivers, both major and minor, that may influence these signals. In the text passage cited by the reviewer (now L184-186), we refer to the methods section, where the potential contribution of other sources is mentioned, and where the reader is directed to Suppl. Text 2.2 for a more detailed discussion on this subject.

L189-194: This makes it seem as though Paffrath's dataset is one example among many. Aren't there just two Nd datasets-- yours and hers?

No longer relevant as text was changed.

L195: Why 'preceded'? Is there some temporal ordering implied here?

Yes, there is a temporal ordering implied, as explained in the section following this statement (now L203-233). However, since this sentence appears earlier and we want to avoid confusion, we have adjusted the wording to reflect a mixing relationship rather than a temporal component. L191-192.

L203: In earlier studies, there is a small, but always measurable, fraction of sea-ice melt below the halocline. Is that not present in your data?

We can confirm the presence of up to 2% of meltwater at these depths (100-200m). We have added this information to the main text along with the observation of a brine signal in the upper 100 m (L154-159). We also adjusted the quoted sentence. L196-200.

L212: I don't think absorption is an appropriate word to use here. "Mixing" maybe.

This terminology was adopted from Bert Rudels, but we have now changed it following the suggestion. L205.

L214-221: I get it; but maybe you could explain this more simply. Or maybe start with the observation, and then the explanation.

We have reformulated this passage for the sake of clarity. L207-219.

L234: Figure 3 makes it look like there is no Nd data between the end of November and the end of January. If this is the case, can this time period be described?

While no ϵ_{Nd} data are available for this period due to the extremely difficult winter sampling conditions, the consistency of the f_{RIV} distribution derived from the extensive $\delta^{18}O$ dataset (Fig. 3a) suggests minimal variability beyond what can be resolved by ϵ_{Nd} data before and after this period. i.e. the freshwater lens observed during this period is unlikely derived from different sources. Furthermore, the sea ice ϵ_{Nd} profile supports this conclusion, at least for the mixed surface layer, demonstrating the continuity of the ϵ_{Nd} signatures during this period.

L240: "explained" seems like the wrong word here.

No longer relevant as text was changed.

L242-245: I don't understand the binary set up in this paragraph. It seems to me that the ice is obviously drifting over different surface waters. This has been known for a very long time ... the speeds and directions have long been measured to be different. I don't see how "drift dynamics" could influence the surface water mass decompositions. These must be water-mass composition changes in the surface ocean. If you're arguing that the shifts take place primarily on the shelves ... and not over the pelagic central Arctic ... then I don't see that you've made your argument. In the study we have not seen data on either the Siberian wind shifts or open Arctic basin wind shifts (both of which we know can redirect the surface plumes.) Maybe I'm just missing something. If so ... what?

This appears to be a misunderstanding due to the vague wording in our original manuscript. We referred specifically to the sampling setup during MOSAiC, where ice drift, unlike ship-based sampling, was non-linear and influenced by changing winds. This movement made continuous, linear sampling impossible and led us to exclude inertial oscillation—the circular motion of sea ice due to the Coriolis effect—as a significant factor in water column variability. The ice floe's movement can create patterns in 2D section plots that may not accurately represent the water column, as it can cause repeated or displaced features from crossing the same water mass multiple times. Thus, we are not suggesting that drift dynamics change surface water composition (in fact, we show the opposite in our ms). Instead, we emphasize that the drift dynamics can lead to apparent shifts in water mass composition in the plots, which do not reflect actual changes. We have clarified this in the revised text, noting that ice drift had only a minor effect on observed tracer variability, given the observed sea ice dynamics and the low spatial and temporal resolution of our data. Even the high-resolution river water fraction distribution from the $\delta^{18}O$ dataset (Fig. 3a) shows no small-scale variations beyond general water mass advection patterns. L235-240.

L256-258: Mechanism for and cause of what?

Rephrased to improve clarity. L290-292.

L258: Could rephrase for clarity. Were the samples collected in these intervals, or is the discussion herein using these intervals, and are they selected for a specific reason or are they arbitrary intervals?

Rephrased for clarity. The intervals were identified based on the specific trends observed in the sea ice profiles. L292-295.

L260: the antecedent to "this" is unclear.

Changed for clarity. L294.

L262: Agreed, for Eps_Nd. But for bulk Nd, the deviations don't appear so minor to me.

In Suppl. Text 3, which we refer to in the main text, we mention that an upward shift of [Nd] by 10 cm would increase the correlation coefficient between [Nd] and $\delta^{18}O$ from 0.4 to 0.8, which is comparable to the effect of the salinity correction. Although the [Nd] shift may appear larger, it is therefore of a similar order of magnitude to that of salinity or ϵ_{Nd} .

L263: Which parameter do you expect to be more permeable and why (citations could help).

The original text was misleading, as it implied a comparison between $\delta^{18}\text{O}$ and other parameters concerning their sensitivity to changes in permeability. It is evident that brine-associated parameters exhibit greater sensitivity, given that most of the $\delta^{18}\text{O}$ originates from the ice crystal lattice, as stated in the main text. While we could theoretically compare the sensitivities of salinity and REEs, including Nd, the lack of relevant studies on REEs precludes us from doing so. Therefore, the revised text now emphasizes the overall sensitivity of brine-associated parameters to variations in permeability. L297-302.

L265: either use this consistently or just say 'water' here.

No longer relevant as text was changed.

L263-267: It would help the reader parse this information if it were two separate sentences. As-is, the interjection about $\delta^{18}\text{O}$ pulls the reader one way; then it is surprising to be talking about concentrations at the end and not isotopes.

We rephrased this passage for clarity. L297-302.

L269: The authors do a great job mitigating brine spillage; especially in the processing phase. However, in the methods, there is no statement regarding spillage during core extraction; which unfortunately can't be quantified but is likely the moment with the highest amount of exchange.

We have added a statement to the methods section acknowledging the unavoidable brine spillage that occurs during core extraction, L484-486. As stated in the main text and described further below, while our approach cannot eliminate the effects of spillage, it can minimize them. L302-305.

L274-276: Fine to crow a bit ... but perhaps it's more important to hammer home a message about the small-scale (intra-flow) variability in porosity changes, which researchers thinking about sea ice-coring need to be cognizant of.

We welcome this suggestion and have modified the revised version accordingly. L311-314.

L276-278: Did they see a different pattern, did their study not discuss salinity? Did they mis-interpret the salinity profile?

We now briefly mention the results of this study concerning salinity distribution, which was essentially uniform at the time of our sampling and did not reflect the trends we observed. L309-311.

L282: What parameters were continuously measured? It looks like the parameters ($\delta^{18}\text{O}$ and Nd/epsilon-Nd) were measured routinely, but not continuously.

Changed to 'regular sampling' as ϵ_{Nd} samples were taken every 1-2 months and $\delta^{18}\text{O}$ samples almost every week. L317.

L283-284: Yeah; this is really cool.

We wholeheartedly agree! The time and effort invested in obtaining the data for this figure were immense, but it was obviously worth it!

L300-303: This sentence is hard to understand.

We have rephrased the passage to enhance clarity and included additional details to improve understanding. L335-341.

L314: It's not obvious to me what this has to do with the continuity of the profile??

What we meant was that constant sea ice profiles without changes would not have allowed such a conclusion. We have now removed this point because the consistent incorporation from the surface water is obvious from the seawater-sea ice comparison.

L332: does this mean that your cores did not have a shift -- or that the shift, also observed in your cores, contradicts the epsilon,nd signature. Could this be floe-based variability? Or could both be true since the surface formed before sampling began?

We have clarified this point in the text. Our cores did not exhibit the observed shift. We now argue that this is floe-based variability established during thaw-freeze cycles in early winter, as only a few winter ice cores in the $\delta^{18}\text{O}$ timeseries (Mellat et al., 2024) show depletion in the uppermost 20 cm. L361-368.

L337: HREE/LREE - I didn't see these terms spelled out above.

We introduced the abbreviations and the ratio in the *original* manuscript at L182, and it is now used frequently in that section (L224-233), so it does not need to be reintroduced later.

L342: This is an intriguing hypothesis. But it's just sort of hanging there. What complexes? And how would these impact fractionation between ice and water? And are there observations in the Laptev and Kara seas or the deltas that might corroborate the suggestion. I would want something more ... or else clarify that this is just a guess.

We clarified that this is just a guess, as there are currently no data available on REE complexation in Arctic rivers. L374-377.

L343-344: This is a vague phrase; not sure what mean.
No longer relevant as text was changed.

L346-347: "disparities of the complexation behavior": are you talking about links to the water molecules ... or to other dissolved things? Or do you know?

We now clarified in the revised text that we referred to REE complexation by inorganic and organic ligands, which could not only account for differential incorporation of Nd sourced from different rivers, but also lead to preferential HREE rejection compared to LREEs (HREE are almost entirely bound in stable carbonate complexes). L377-381.

L355: 'precision' seems odd here. Do you mean that the differences are small compared to the precision of your measurements? If not, then maybe clarify.
This is no longer relevant as we have reworded that section.

L358: I think this overstates what's been accomplished, at least to this point in the manuscript.
This is no longer relevant as we have reworded that section.

L362-366: The structure of the sentence doesn't make sense. The first half implies that there should be significant variation in distributions ... the second half notes that it has been found--then why "Despite"... ? Do you mean something like, "Despite the variability of their distribution has been tracked to a relatively simple dynamical mechanism."
We apologize for this incorrect wording. This passage has been removed from the manuscript.

L366-368: I really like this narrative; and I suspect you are spot on. But I don't think you've quite made the case above. You seem to assume a "freight train" model of the TPD, in which parcels are loaded on at the Siberian shelf break, and no one gets on or off along the trek to Fram Straight. If that's your image, I think it needs to be more explicit, and perhaps an argument made for it. Alternatively, if your view is that the Ob/Yenesei and Lena river plumes enter the TPD at different points, then what is the evidence for temporal variability? Couldn't this just be the floe crossing over distinct, but relatively stable, spreading plumes?
We now clarified our position in the last paragraph of the section "Origin, distribution and variability of Siberian fresh water along the Transpolar Drift". L270-288.

L368: Does this mean: the heterogeneity in observations along the TPD? Why does it imply shelf hydrography processes rather than variability in basin circulation.
Yes, that is correct. We were referring to the heterogeneity in observations along the TPD. We have improved the wording for clarity and now also address the limited impact of basin circulation in the revised manuscript. L277-281.

L377-378: A couple of thoughts:

1) cool

2) did you determine this in this study, is there a citation from another mosaic study?

3) adapts is a bit of an odd word. is replaced by?

Maybe: The sea ice drift encountered different water masses every 14 days"

Also, do other mosaic studies comment on the encountering of multiple water masses (i.e., lagrangian sea ice drift, but not a lagrangian ocean drift) yet? Katrin Schmidts paper alludes to it. Did any of the overview papers call it out?

1) Yes, it is!

2) We determined this number based on our data, now also clarified in the main text. L398-400.

3) No longer relevant as text was changed.

4) Rabe et al. 2022 and Schulz et al. 2024 reported surface salinity changes and generally attributed those to ocean surface variability and seasonal stratification changes without providing data that resolve source contributions.

L378-380: In my experience, most researchers assume that there is significant divergence between the water and ice motions--in part based on data from previous drift expeditions ... and more recently from profiling ice buoys. Where I've seen people assume coherence it has been for lack of detailed data on the surface water velocities.

We agree and have removed this narrative from our manuscript, see response to main comment on this issue.

L382: rapid is a relative term; could you constrain this a bit? "...the redistribution of sib fw over an annual cycle..."
Adjusted as suggested. L396.

L386-388: Yes. But also: even in the absence of short-term variations, the ice and the surface water would diverge, as the ice "should" (per our basic Ekman theory) travel both faster and at an angle to the underlying water.

We agree. We removed this argumentation from the manuscript, see response to main comment on this issue.

L393: the release of ...material from sea ice...

No longer relevant as text was changed.

L394: This is a bit confusing -- could you quantify how much the release of sea ice materials influenced the water column and how much of the water column is influenced by advection of Atlantic-water?

We addressed this issue in our response to the related main comment. Please see above.

L397-398: I'm not sure that "neglected" is a fair characterization to the growing body of literature working on this question. For example, Damm et al, 2015 poses the notion that dissolved gasses in central Arctic Ocean might be transported from the siberian margin and later (2018 I think) demonstrate it. In 1990s there was a paper regarding the question of "does ice transport pollutants. This question has been on the radar, and this study adds important information to that question.

As we clarified in our response to your general comment, the original text was misleading. Our intention was to highlight that the specific role of sea ice formation *along* the entire TPD in the central Arctic Ocean has been largely overlooked, not the broader research question itself.

L461-463: It's unclear which part of the approach was adopted for these purposes.

We meant the combination of depth intervals from individual ice cores. We have adjusted the text accordingly. L481-483.

A few thoughts/questions:

brine spillage during sampling was likely at a maximum when the core was pulled from the corer and placed in the bag.

This is correct. However, while the new approach cannot eliminate the effects of brine spillage during core retrieval, it minimizes them. Our data show that taking a larger number of ice cores increases the fraction of trapped brines compared to spilled brines, minimizing the impact on the tracers associated with the brine. We have added this information to the method section and modified Suppl. Text 3 accordingly.

Is the wash with DI water a method that's been used before for trace metals in sea ice? Could you provide a citation? Most trace metal work with sea ice shaves the edges of the ice with a titanium blade. Has the water method been shown to reduce contaminants from the coring process?

While physically removing the outer ice layer is generally effective in eliminating surface contaminants, it can sometimes cause their redistribution if ice fragments are displaced during scraping. By rinsing with MilliQ water prior to scraping, we can remove most surface contaminants, minimizing the risk of redistribution. This decontamination approach has been used in the preparation of glacial ice cores (e.g., Aarons et al., 2016, *EPSL*, <https://doi.org/10.1016/j.epsl.2016.03.035>) and is now applied by others without the ether step for trace metal and isotope analysis (Choi et al., *in prep.*). Notably, this rinse does not affect the brine content of the ice core, as only enclosed brines remain after sampling, and these are not influenced by brief exposure to MilliQ water. We have included this clarification in the text, L472-476.

Did your brine spillage control include the wash step, in the next step you state "to serve as a reference for the latter two processes" but the reader has to do a lot of re-reading to figure out what the latter two processes were.

Yes, the control included the wash step. We acknowledge the lack of clarity in the original text and have revised it accordingly. Our intent was to refer to sea ice heterogeneity and brine spillage during sampling, not the wash step. L487-488.

L490: I don't know this method off the top of my head and would have to go to the citation, but just to double check: Is it correct that the solution was washed with di-ethyl ether after it was redissolved in 6M HCl? Does it reprecipitate?

Yes, your description is correct, but no reprecipitation is needed. After redissolving the sample in 6M HCl, the di-ethyl ether is added to perform solvent extraction. The ether forms a complex with Fe, which is less dense and thus separates from the acidic phase. This process effectively removes Fe from the sample, while the trace metals remain in the acidic phase. The ether extraction is repeated to ensure that most of the residual Fe is removed. We acknowledge that the text was unclear and have now improved it. L513-515.

L595-596: What about the uncertainties from scatter in the end-members? In my experience, those usually overwhelm the analytic uncertainties. (As an example, the Rosen, et al., 2015 paper estimates a 10% error on sea-ice formation as a result of uncertainty on the end-member value of 18O in Siberian river waters.)

We have performed a Monte Carlo simulation to account for scatter in the endmembers. Please see our reply to your general comment above.

L620: What does this mean for how you interpret your data?

We have clarified in the revised text that this uncertainty is reflected in all relevant figures and is factored into our interpretation of the data. L659-662.

L623: Different citation format

Resolved.

L634-639: Fair enough; your choice. But why did you use this 2-step method, rather than define a 4-equation/4-end-member problem, and solve it in one matrix inversion?

We have adjusted the calculations as suggested. Please see our reply to your general comment above and to the general comment of Reviewer 2.

Figure 1: Flip the color bar, up/down.

Done. We have also updated the color scheme for the timeline in Figs 1, 5 and 6 to ensure it is distinct from the scheme used for tracer data in the other figures. This change is intended to avoid confusion and clearly differentiate metadata from the actual tracer data presented in the remaining figures.

Figure 3: The figure labels are awfully small.

We agree and have enlarged the figure labels.

L705-707: Should figs b-e be gridded? The data is well spaced out and, given the high variability in conditions that the authors describe, does the gridding permit over-interpretation?

The four sections illustrate the distribution of Nd isotopes and the parameters derived from them. Nd isotope signatures originate from large-scale geological formations and broadly reflect water masses in the oceans on a basin or inter-basin scale, rather than capturing small-scale processes. Each water mass carries a distinct *regional* signature that remains stable over distances of several hundred kilometers. The variability we describe here pertains to differences in water mass distribution, not fine-scale processes, which, although significant, are beyond the scope of this analysis. A higher resolution of the water mass distribution will, however, unlikely be achieved by a higher number of Nd isotope data points. For instance, in the Fram Strait in 2012 (Laukert et al., 2017, *GCA*), only five distinct water masses were identified despite high spatial resolution and over 100 Nd isotope data points. Similarly, in the Laptev Sea, only three water masses were detected with comparable resolution (Laukert et al., 2017, *EPSL*). Neither study detected small-scale processes or circulation features using Nd isotopes. Therefore, increasing Nd isotope sampling density in the TPD is unlikely to reveal additional distinct water masses or finer-scale details, nor significantly alter the overall pattern in the gridded distribution.

The high resolution of the water mass distribution is best illustrated by the extensive O isotope dataset, as shown by the river water fraction in Figure 3a. We assume that the neodymium isotope distribution at a finer resolution follows the patterns of the O isotope distribution, given that both parameters are strongly linked, allowing readers to infer the details of the neodymium distribution from the river water distribution derived from O isotopes. Consequently, we continue to use the gridded representation to emphasize the relative distribution of water masses. Individually colored data points would not effectively convey the broader patterns within the system.

Figure 5: This chart was hard to read, would it help to add a time component (z-axis) to the seawater data? It might help to remind the reader how the ice data was determined (i.e., is this just one core or a composite of several cores/dates).

L739-740: It took way too long to understand this chart. Needs a better explanation. Imagine a reader that does not know what you've done here ... and explain it clearly.

We have added a time component to the seawater data in Figs 5 and 6 to enhance clarity. Additionally, we have refined the figure captions to provide a clearer explanation of the plots. Please note that the distribution of all tracers has slightly changed due to a refinement of the ice age model. Furthermore, the $\delta^{18}\text{O}$ dataset has been updated and quality-checked based on duplicate samples, which has led to changes in the $\delta^{18}\text{O}$ data and parameters derived from them in Figs 2-6.

Figure 7/ L787-788: I really did not get this image from reading your text. I think you need to state this early and clearly in the text. There's a lot of discussion of the basic fact that the ice and surface water diverge--"are not aligned"--which is not nearly clear enough to distinguish between your "established" and "emerging" views. Your take-home message in this chart has to do with the impact of short-term variations (the 'eddy term' in classical phys oce texts) vs the long-term mean flow. But you don't make that case clearly in the text.

We have removed the "decoupling" argument as a central point distinguishing between established and emerging views. Instead, we now focus on the role of shelf hydrography as the primary driver of TPD variability, and ice-ocean exchange *along* the TPD, rather than basin circulation or short-term features.

L789: The meanings of the various lower case and capital letters need more explanation. It may help to move away from "a" and "b" as well since those are the indications for the sub-figures.

We changed the letters and now explain lower case and capital letters in the figure caption.

Reviewer #4:

We are very grateful for the contribution of this reviewer and fully support the initiative to provide a direct opportunity for Early Career Researchers for training in the peer review process.

Editorial remarks (by line #):

Line 1: Is "material" a better word than "matter"? I found the phrase "Siberian matter" a bit strange.

We appreciate the suggestion and gave this careful consideration. We prefer the term 'matter' for two main reasons. First, 'matter' is a well-established term in oceanography, as seen in phrases like 'organic matter' and 'suspended particulate matter,' and was similarly used in this context by Krumpal et al. (2019, *Sci. Rep.*) in discussing matter transport via sea ice along the TPD. Second, 'matter,' by definition, encompasses anything with mass that occupies space, including atoms, molecules, elements, and compounds, while 'material' more commonly refers to substances used to construct objects (e.g., wood, metal, plastic). Thus, 'matter' better represents the broad range of physical substances involved in our study. For these reasons, we believe 'matter' is the most accurate and consistent choice for our study.

Line 69: Ditto. It's a bit weird to use "matter fluxes" here. Do the authors think other phrases such as "material fluxes" or "elemental fluxes" work better? If so, the authors should consider changing this word/phrase throughout the manuscript. While we understand the concern, we believe that 'matter fluxes' is the most suitable term in this context. Unfortunately, alternatives like 'elemental fluxes' are too restrictive, as they would exclude the transport of non-elemental substances such as microplastics. Given the broad range of substances covered by our conclusions, including trace metals, particles, nutrients and contaminants, 'matter fluxes' remains the most accurate term.

Line 90: I don't think "the reasons mentioned above" are immediately before this sentence or even in this paragraph. If that's the case, the authors should consider summarizing the reasons briefly and adding them here.

This is no longer relevant as we have reworded that section.

Line 614: I may have missed it, but what is "CO"? Please clarify.

We meant "Central Observatory", the abbreviation of which was introduced at the beginning of the method section. This abbreviation was only used twice, and we decided to remove it from the text for clarity.

Lines 671–673: The end of the first drift is not labeled by a dashed black circle.

Our previous wording was unclear. We meant that only the FYI formation was marked with a circle. We have updated the figure caption of Fig.1 to reflect this more accurately. We have also updated the figure by adding the new $\delta^{18}\text{O}$ stations and using a distinct color scheme for the timeline, different from the one used for the tracer data in the other figures. The latter change is intended to avoid confusion and clearly differentiate metadata (shown in Fig. 1) from the actual tracer data presented in the remaining figures.

Lines 675–678: What are those dashed arrows? Please clarify.

We have clarified that these arrows indicate the approximate advection paths of the river inputs. L706-708.

RESPONSES TO REVIEWER COMMENTS IN THE SECOND REVISION ROUND

Line numbers correspond to the version with marked edits.

Reviewer #1:

I appreciate the efforts taken by the authors to address the comments of their initial reviewers. I think the ms. has improved significantly from the first version. This new version reads well and is superbly illustrated. I still hold reservations on many aspects of the work, but their arguments are generally all valid. While my original sentiment holds true – that these are interesting data collected from a unique sampling campaign – I still have two reservations.

We thank the reviewer for their second review and thoughtful feedback. We are encouraged to hear that the re-write has addressed many of the initial concerns and that the manuscript is now perceived as significantly improved.

The first is a technical issue: The samples were collected in a lagrangian sense (drifting with the floes), but the data seems to be interpreted in an eulerian sense. I worry about regional interpretations being made from locally collected data in this instance – e.g., how much of their sampling could be simply caused by local ice floe-sea water interaction/evolution versus regional inputs/outputs?

We appreciate the reviewer's insightful comment. However, based on the observations presented in our study and the underlying physical processes, we are confident that the data reflect regional inputs and outputs rather than being dominated by local ice floe-sea water interaction or evolution. Below, we outline two key arguments supporting this interpretation:

1. If the observed tracer signatures were solely a result of local ice floe-sea water interactions, they would remain largely unchanged during drift, as the water column would drift alongside the ice. However, our study reveals *highly variable signatures of Siberian origin at the ice-water interface*, which are inconsistent with a purely local interaction mechanism.
2. Tracer signatures at the ice-water interface are consistent with those observed at depths of 50 to 100 meters (Fig. 3a). This consistency indicates that the signatures extend *throughout the mixed layer and into the halocline*, where ice-ocean drag and Ekman transport are negligible. As such, the observed patterns are not artifacts of the unique sampling setup but instead reflect large-scale upper ocean circulation and its integration into sea ice, primarily influenced by the variability in source regions of the water on the shelves.

In recognition of the need for clarification, we have added a sentence to the manuscript, L255-258:

*The river water distribution along the TPD during the MOSAiC expedition reveals notable spatiotemporal variability in advection. Variations in ice drift direction caused by inertial oscillations⁵⁴—the circular movement of sea ice driven by the Coriolis effect—appear to have no impact on tracer-derived fraction distributions when plotted over time and thus following ice motion, likely due to the low spatial resolution of the tracer data. Even in the well-resolved river water distribution, only broad advection and mixing patterns are visible, while smaller features fall below data resolution (Fig. 3a). **These patterns also remained unaffected by local ice-water interactions during the drift, as tracer-derived fractions are consistent throughout the upper water column, extending to depths of up to 100 meters—well beyond the direct influence of ice-ocean drag and Ekman transport.** Throughout the campaign, sea ice drifted up to six times faster than surface waters³³, moving over high-fRIV waters toward the Fram Strait until mid-March 2020, when the drift shifted towards the Nansen Basin. Therefore, tracer changes observed until spring 2020 mainly reflect variability in river water advection along the freshwater-rich TPD, along with the high-fRIV waters found later that year near the North Pole and at Fram Strait.*

This clarification highlights how the river water distribution reflects broader regional processes, further supporting the validity of our interpretation. We trust that this addition addresses the reviewer's concern effectively.

My second issue is simply that I was uninspired by their conclusions (a problem I had in the original version). This may not be an issue for publication, as the other reviewers seemed to consider this work important. I defer to the editors and other people more interested in the themes addressed in this work; is this interesting work?

Considering the current challenges in Arctic research, we believe this work marks a significant advancement in understanding matter dispersal in this rapidly evolving region. The geopolitical constraints on accessing Siberia underscore the importance of innovative offshore, tracer-based methods like those utilized here to reveal dispersal pathways. These approaches are crucial for evaluating the spread of hazardous substances, both now and in the future, with far-reaching implications for biogeochemistry, ecosystems, and human health.

In summary, I think the re-write has helped tremendously, and I could envision this published in Nat. Comms.. Any perceived ambivalence to publication should not be seen as a demerit, rather my own ignorance of the greater science this work addresses..

We value the reviewer's insights and the broader discussion they have prompted and hope that our response clarifies the study's importance.

Reviewer #2:

The authors have fully addressed my questions and done a good job responding to the points brought up by other reviewers. I am satisfied with the authors' explanations and edits in the revised manuscript. I recommend accepting the manuscript.

We thank the reviewer for their positive evaluation and recommendation to accept the manuscript.

Reviewer #3:

This is a resubmission of a manuscript that I reviewed previously.

All of my comments and criticisms have been fully addressed. In fact, the authors' responses were more detailed and extensive than usual, and so have made interesting reading in themselves.

They have extensively reworked several parts of their argumentation; and they have added significantly to their supplementary material. Changes to the charts and wordings of various sentences and paragraphs have made the manuscript much clearer and easier to read.

I recommend publication of the current manuscript without further revision.

We sincerely thank the reviewer for their feedback and kind words regarding our revisions. We are pleased that our efforts to address the comments and enhance the manuscript were appreciated.

Reviewer #4:

Again, we are very grateful for the contribution of this reviewer and fully support the initiative to provide a direct opportunity for Early Career Researchers for training in the peer review process.